# Learning improves decoding of odor identity with phase-referenced oscillations in the olfactory bulb

Justin Losacco[1,2†‡], Daniel Ramirez-Gordillo[2†], Jesse Gilmer[1,3], Diego Restrepo[1,2*]

[1]Neuroscience Graduate Program, University of Colorado Anschutz Medical Campus, Aurora, United States; [2]Department of Cell and Developmental Biology, University of Colorado Anschutz Medical Campus, Aurora, United States; [3]Department of Physiology and Biophysics, University of Colorado Anschutz Medical Campus, Aurora, United States

**Abstract** Local field potential oscillations reflect temporally coordinated neuronal ensembles—coupling distant brain regions, gating processing windows, and providing a reference for spike timing-based codes. In phase amplitude coupling (PAC), the amplitude of the envelope of a faster oscillation is larger within a phase window of a slower carrier wave. Here, we characterized PAC, and the related theta phase-referenced high gamma and beta power (PRP), in the olfactory bulb of mice learning to discriminate odorants. PAC changes throughout learning, and odorant-elicited changes in PRP increase for rewarded and decrease for unrewarded odorants. Contextual odorant identity (is the odorant rewarded?) can be decoded from peak PRP in animals proficient in odorant discrimination, but not in naïve mice. As the animal learns to discriminate the odorants the dimensionality of PRP decreases. Therefore, modulation of phase-referenced chunking of information in the course of learning plays a role in early sensory processing in olfaction.

*For correspondence:
diego.restrepo@cuanschutz.edu

†These authors contributed equally to this work

‡Deceased

**Competing interests:** The authors declare that no competing interests exist.

## Introduction

Animals must modulate early sensory processing to optimize navigation of their environment (*Baker et al., 2018*; *Pakan et al., 2018*). This experience-dependent shaping involves an interplay between sensory input, behavioral state, arousal and motor activity. Neuronal activity, temporally organized by local field potential (LFP) oscillations, carries distinct information at different phases of each cycle (*Jensen, 2001*; *Kepecs et al., 2006*; *Lisman and Jensen, 2013*). For example, in the hippocampus different phases of the theta LFP (2–12 Hz) preferentially encode present and future spatial locations, illustrating temporal 'chunking' of information (*Amemiya and Redish, 2018*; *Siegle and Wilson, 2014*). Place cell preferred fields are encoded temporally through phase precession in relation to hippocampal theta oscillations (*Buzsaki, 2004*; *Lisman, 2005*; *O'Keefe and Recce, 1993*; *Skaggs et al., 1996*). Additionally, LFP oscillations in different frequency bandwidths can be related through cross frequency coupling (CFC), which can further enhance information carrying capacity (*Canolty and Knight, 2010*; *Engel et al., 1999*; *Florin and Baillet, 2015*; *Fries, 2005*).

One example of CFC is PAC, wherein the amplitude of a faster oscillation (e.g. gamma, 35–95 Hz) is related to the phase of a slower oscillation (e.g. theta, 2–12 Hz) (*Chrobak and Buzsáki, 1998*; *Soltesz et al., 1993*). PAC occurs during task engagement in rodent hippocampus (*Bragin et al., 1995*; *Lisman and Idiart, 1995*; *Soltesz et al., 1993*), rodent orbitofrontal cortex (*van Wingerden et al., 2014*), macaque auditory cortex (*Márton et al., 2019*), human hippocampus (*Lega et al., 2016*) and human visual cortex (*Daume et al., 2017*; *Seymour et al., 2017*). Interestingly, *van Wingerden et al. (2014)* described how theta-gamma PAC strengthens in rat orbitofrontal cortex with learning for trials in which the correct decision was made in the olfactory go no-go task.

The study of PAC in the earliest sensory processing brain areas is virtually non-existent. The olfactory bulb (OB) is an ideal location to study whether PAC is relevant to sensory processing; its intrinsic oscillatory activity is linked to odorant signal processing (*Doucette et al., 2011*; *Gschwend et al., 2012*; *Li et al., 2015*; *Smear et al., 2011*). The rhythmic activation of olfactory sensory neurons (OSNs) through breathing generates a theta-frequency (2–12 Hz) respiratory-related LFP in the target region of the OSNs, the OB, that is coherent with oscillations in downstream processing areas such as piriform cortex (PC) and hippocampus (*Gourévitch et al., 2010*; *Heck et al., 2019*; *Nguyen Chi et al., 2016*; *Tort et al., 2018*). Interestingly, PAC is present in human PC, and the respiratory phase for presentation of stimuli modulates the ability to retrieve episodic memory (*Zelano et al., 2016*). Downstream neural networks receiving olfactory information from the OB could be tuned to respond to sensory input at certain phases of this theta LFP (*Buzsáki, 2010*; *Jensen, 2001*) and could send back modulatory signals during other phases (*Kay, 2015*; *Kepecs et al., 2006*).

Decades of work have characterized LFP activity in the OB in various states of wakefulness (*Adrian, 1950*; *Chery et al., 2014*; *Li et al., 2012*) and olfactory task engagement (*Gourévitch et al., 2010*; *Martin and Ravel, 2014*). Surprisingly, this LFP signal changes in response to prior experience (*Courtiol and Wilson, 2017*; *Gire et al., 2013*). Work in the go no-go operant conditioning task where the animal learns to respond to the reinforced odorant for a water reward has revealed broadband LFP power changes for the rewarded versus the unrewarded odorant with discrimination learning (*Beshel et al., 2007*; *Martin et al., 2004*; *Ramirez-Gordillo et al., 2018*; *Stopfer and Laurent, 1999*). Additionally, changes in the balance of sensory and corticofugal inputs to OB interneurons can alter timing of beta and gamma oscillatory states, suggesting that information transfer from the OB is influenced by top-down signaling (*David et al., 2015*). Also, these investigators posited that slower theta oscillations gate temporal windows for sensory and centrifugal inputs to ascend and descend, respectively, suggesting an important role for PAC in gating information transfer. Since PAC exists in the OB in anesthetized (*Buonviso et al., 2003*) and awake, idle rats (*Rojas-Líbano et al., 2014*), we wondered whether PAC changes with learning, reflecting unsupervised Bayesian learning or supervised learning through top-down modulation sensory information (*Hiratani and Latham, 2019*), and whether encoding this information by the power of high frequency oscillations differs as a function of the phase of the theta LFP.

Here, thirsty mice implanted with tetrodes in the OB learned to associate an odorant with water reward in the go no-go task, discriminating between two monomolecular odorants (e.g. between isoamyl acetate vs acetophenone, IAAP) or between a monomolecular odorant and an odorant mixture (between ethyl acetate and a mixture of ethyl acetate and propyl acetate, EAPA). LFP was analyzed over task learning for changes in PAC and for the ability of theta phase-referenced high frequency oscillatory power—termed phase referenced power (PRP) hereafter—to encode for the perceived contextual identity of the odorant.

We demonstrate that the strength of PAC, quantified as the modulation index (*Tort et al., 2010*), changes for the rewarded odorant (S+) as the animals learn. Interestingly, the variance of the angle of theta with the strongest gamma oscillation (peak angle) increases for the unrewarded odorant (S-) over the course of learning. These changes in PAC raised the question whether a downstream observer focused on high frequency oscillations at different theta phases would find a difference in stimulus prediction with learning. We found that the ability to decode stimulus identity by PRP improves with task proficiency and the dimensionality, a measure of separable representation within signals, decreased as the animal learned to differentiate the odorants, suggesting that a stable and unambiguous representation of odor identity arises in PRP after learning.

## Results

The experiments were designed to study encoding of odorant identity by cross-frequency coupling. We report data obtained in two different sets of experiments that we name Exp1 and Exp2. As shown in *Supplementary file 1*-Table S1 the two experiments differed in electrode locations, mouse genotype, device implantation (tetrode vs. optetrode), and odorant pairs used (see also Materials and methods). Mice learned to discriminate between rewarded (S+) and unrewarded odorants (S-) presented in pseudorandomized order in the go no-go olfactory task (*Figure 1A–C*). The dataset is comprised of 119 recording sessions in 30 mice. The odorant pairs tested were different

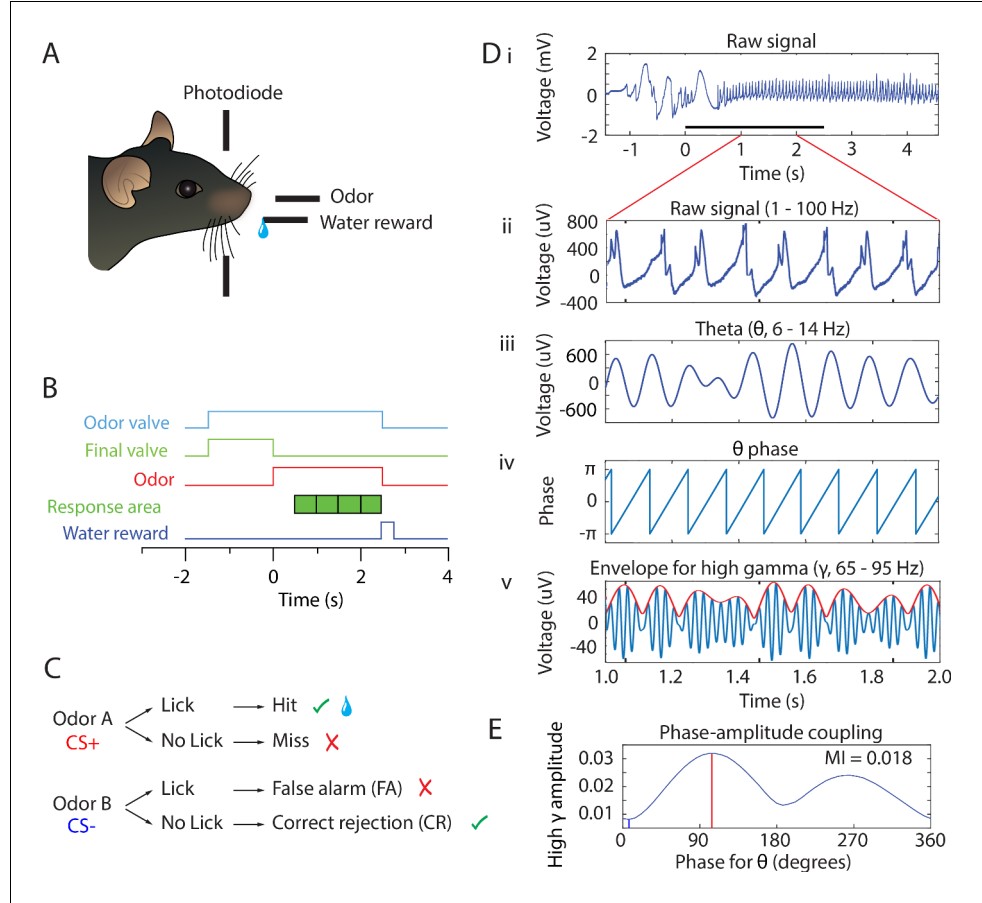

**Figure 1.** Behavioral task and PAC analysis. (**A**) Behavioral apparatus. Mouse self-initiates trials by breaking a photodiode beam. Odorants and water are delivered according to the timeline in B and the decision tree in C. (**B**) Timeline for a single trial. When the animal enters the port the air flow is diverted by turning the final valve output to the exhaust and the odorant valve is turned on. At this time the odor builds up to steady-state concentration flowing away from the animal for 1–1.5 s. At time 0 the final valve air flow is turned towards the odor port for 2.5 s resulting in odor onset 100 ms after the valve is actuated. In order to trigger water reward for the rewarded odorant the mouse must lick at least once during each 0.5 s block for four blocks in the response area. If the stimulus is rewarded and the animal licked appropriately, a water reward is delivered. (**C**) Decision tree for each stimulus. Green check mark means the correct decision was made. Red 'X' mark indicates an incorrect decision. Water reward is represented by the water droplet symbol in the case of a hit. (**D**) PAC data analysis of the LFP. For each electrode, raw signal collected at 20 kHz (i) is bandpass filtered to 1–100 Hz for broadband LFP (ii) or filtered into different frequency bands as needed (e.g. theta 6–14 Hz in iii or high gamma 65–95 Hz in v, blue line). Hilbert transform is used to calculate the theta phase (iv) and the amplitude envelope of higher oscillations such as high gamma (65–95 Hz) (red line in v). Theta phase and the envelope of the amplitude of high gamma are then used to calculate the probability for high gamma amplitude at specific phases of theta (PAC, shown in E). (**E**) Probability for high gamma amplitude at specific phases of theta. In this example, PAC is strongest at ~100° and 270°. The coupling strength is quantified by the modulation index, MI. Peak phase (the phase of theta with the highest amplitude in high gamma) is indicated with a red line and trough phase is indicated with a blue line.

The online version of this article includes the following figure supplement(s) for figure 1:

**Figure supplement 1.** Time course for odorant concentration measured at the odor spout in the olfactometer.

**Figure supplement 2.** Simulation: PRP wavelet analysis for theta-gamma waves with/without PAC.

volatile compounds (or mixtures) whose nomenclature addresses the odorant names and the experimental set (e.g. IAAPexp1, see *Supplementary file 1*-Table S1 for the nomenclature). *Supplementary file 1*-Table S2 enumerates the total number of sessions per odorant pair, mouse, and experiment.

In the go-no go task mice start the trial spontaneously by poking their nose into the odor spout and licking on the lick port. The odorant is delivered at a random time 1–1.5 s after nose poke (the time course for odorant concentration is shown in *Figure 1—figure supplement 1*). The mice must decide to either lick a waterspout at least once during each 0.5 s bin in the 2 s response area (green blocks in *Figure 1B*) to obtain a water reward for the rewarded odorant or refrain from licking for the unrewarded odorant (*Figure 1C*). We did not punish for licking for the unrewarded odorant, and mice refrain because of the effort it takes to lick. Behavioral performance was termed naïve or proficient when their performance estimated in a 20-trial window was ≤65% for naïve and ≥80% for proficient. After three 20-trial blocks of proficient performance, the session was ended; valence was reversed the next day or another odorant was tested. We recorded the LFP using four tetrodes (16 electrodes) implanted in the OB, and we analyzed the data to determine whether information carried by cross-frequency coupling can encode for the contextual identity of the odorant for naïve or proficient mice. Importantly, reversal experiments from previous studies where we switched the rewarded and unrewarded odorants have shown that the power of the LFP in the OB and mitral/tufted cell spiking encodes for the *contextual* identity of the odorant (is the odorant rewarded?) (*Doucette et al., 2011*; *Doucette and Restrepo, 2008*; *Li et al., 2015*; *Ramirez-Gordillo et al., 2018*). Therefore, when we refer to identity in this publication we do not mean the chemical identity of the odorant, we mean whether the odorant is rewarded or unrewarded. Finally, to evaluate the statistical significance of differences in oscillatory parameters estimated in this study the estimates were either averaged in the time period of odorant application (0.5 to 2.5 s after diverting the odorant to the odor delivery spout) or the statistical significance was evaluated for the entire time course with time points every 0.1 s using a generalized linear model (GLM, see Materials and methods). Furthermore, we complement testing of statistical significance using p values with estimation of bootstrapped confidence intervals (*Halsey et al., 2015*).

## Phase amplitude coupling analysis of the LFP recorded in the go no-go behavioral task

In order to understand whether odorant identity is encoded by cross-frequency coupling in the OB we characterized PAC, which is a cross-frequency coupling mechanism where high frequency oscillation bursts take place at specific phases of low frequency theta oscillations (*Tort et al., 2010*). PAC has been reported in the OB (*Buonviso et al., 2003*; *Rojas-Líbano et al., 2014*), but has not been thoroughly characterized. *Figure 1D–E* show our approach to quantify the strength of PAC using the modulation index, a measure of how localized high frequency firing is within the phase of theta oscillations. *Figure 1Di and ii* show an example of the extracellular LFP sampled at 20 kHz and filtered between 1–750 Hz. The raw signal (*Figure 1Di*) was filtered with a 20[th] order Butterworth filter into different LFP frequency bands (*Figure 1Dii, iii and v*) (theta, 6–14 Hz, adapted from *Nguyen Chi et al. (2016)*; beta, 15–30 Hz; or high gamma, 65–95 Hz). *Figure 1Diii and 1DV* show that the filtered theta and high gamma LFP appear to change amplitude in a coordinated manner. We used PAC analysis (*Tort et al., 2010*) to evaluate the degree of coupling of the amplitude of the envelope of the beta or high gamma LFP on the phase of the theta LFP. *Figure 1Div* shows the theta phase and the red line in *Figure 1Dv* shows the envelope for the amplitude of the high gamma LFP, both calculated with the Hilbert transform as detailed in *Tort et al. (2010)*. *Figure 1E* shows the phase amplitude plot of the normalized distribution of high gamma amplitudes in different theta phase bins. For this example there are two maxima, and the vertical red line corresponds to the phase of theta at which the gamma amplitude is highest (at ~95 degrees); this is called the 'peak phase'. Similarly, the phase with the smallest normalized amplitude is called the 'trough phase' (vertical blue line at ~5 degrees). The strength of PAC was estimated as the modulation index (MI), a measure computed as the Kullback–Leibler (KL) distance to the uniform phase distribution normalized to range from 0 (uniform distribution) to 1 (sharp gamma bursts at a specific theta phase) (*Tort et al., 2010*). The MI for the distribution in *Figure 1E* is 0.018. *Figure 1—figure supplement 2* shows the LFP power spectrogram for simulated data with low gamma oscillations that are either uniformly distributed in the theta phase (MI = 0.0004, *Figure 1—figure supplement 2A*) or with gamma bursts at 180 degrees of the theta oscillation (MI = 0.01, *Figure 1—figure supplement 2B*). Recordings in hippocampus and cortex yield MI values in the range of 0.005–0.03.

## Peak angle variance for Theta/high gamma phase amplitude coupling increases for S- when the animals learn to differentiate odorants in the go-no go task

We proceeded to ask whether the strength of PAC, quantified by MI, and the peak angle change as the animal learns to differentiate odorants in the go-no go task. *Figure 2A–F* illustrates theta/high gamma PAC for all S+ and S- odorant trials in one session for a mouse proficient in differentiating the odorants. The phase amplitude plots for all the trials are shown in pseudocolor in *Figure 2A*, behavioral performance is shown in *Figure 2B*, and the corresponding normalized phase amplitude plots are shown in *Figure 2C*, while the mean average theta LFP is shown for one cycle in *Figure 2D*. There is clear (biphasic) theta phase coupling for high gamma amplitude with maxima at ~120° and 270° for S+ (*Figure 2Ai* and red line in 2C). This strong PAC is reflected by an MI with a mean of 0.02 with small trial to trial variance (*Figure 2E*, red circles), and small variation in peak phase (*Figure 2F*, red polar histogram). In contrast, PAC displays variable strength and phase for S- in this session (*Figure 2Aii*, blue line in 2E and blue polar histogram in 2F). For S- the average MI is smaller and displays large variance ranging from 0 to 0.06 (mean MI 0.014, *Figure 2E*, blue circles). The peak angle for S+ takes place slightly above 90 degrees and varies significantly for S- (*Figure 2F*). The MI and peak angle variance differ significantly between S+ and S- (MI: ranksum p value < 0.001, peak angle variance: ranksum p value < 0.05, n = 50 S- trials, 49 S+ trials).

Browsing through PAC data for different sessions, plotted as in the example shown in *Figure 2A–F*, raised the question whether PAC changes over the course of learning in the go-no go olfactory discrimination task—as reported by mean MI and peak angle variance. We noticed that in sessions where the mouse learned to differentiate the odorants the phase of the peak angle became highly variable when the animal became proficient. An example is shown in *Figure 2—figure supplement 1* where a mouse started the session performing close to 50% correct, and became proficient during the last portion of the session. The phase of the peak angle is highly variable for the last 20 S- trials. We proceeded to estimate mean MI and peak angle variance for all mice. The mean MI and peak angle variance was estimated as the mean in all trials within either the naïve or proficient periods for S+ and S-, calculated between 0 and 2.5 s after odorant application. *Figure 2G* illustrates cumulative histograms for the mean MI (i,iii) and peak angle variance (ii,iv) for theta/beta (i,ii) and theta/high gamma PAC (iii,iv) for the APEBexp1 odorant for S+ and S- odorants for trial ranges when the mice were naïve or proficient. The cumulative histograms for mean MI measured per electrode (*Figure 2Gi, iii*) show small yet statistically significant changes in PAC strength for both the theta/beta and theta/high gamma as the animal learns to discriminate between odorants (GLM analysis, indicates that the differences in mean MI in terms of proficiency and the event type are significant p<0.001, 896 degrees of freedom, d.f., for theta/beta and theta/high gamma, 16 electrodes per mouse, 14 mice). In addition, the circles in *Figure 2Gi, iii* show the per mouse mean MI. A GLM analysis did not yield significant changes for the per mouse average MI as a function of either event type or proficiency for theta/beta PAC (p>0.05, 52 d.f., 14 mice). Per mouse average MI was significant for both event type and proficiency for theta/high gamma PAC (p<0.05, 52 d.f., 14 mice). Finally, for the APEBexp1 odorant pair we found a substantial decrease in peak angle variance for S+ and an increase in peak angle variance for S- for both the theta/beta and theta/high gamma PAC (*Figure 2Gii, iv* GLM analysis for changes in peak angle variance for both proficiency and event type yield p<0.001, 892 d.f. for both theta/beta and theta/high gamma, 16 electrodes per mouse, 14 mice). The changes were also statistically significant when a GLM was computed for the average peak angle variance per mouse (circles in *Figure 2Gii, iv*, GLM p value < 0.01 for theta/beta and p<0.001 for theta/high gamma for both event type and proficiency with 52 d.f.,14 mice).

Similar differences between S+ and S- and changes when the animal became proficient were found for peak angle variance for the other odorant pairs (Supplemental Information and *Figure 2— figure supplement 2*). Furthermore, we evaluated the changes in PAC strength quantified as MI and peak angle variance by analyzing per-experiment mean values (averaged over odorant pairs, *Figure 2H*). GLM yielded significant p values for peak angle variance for theta/high gamma PAC for S+ vs. S- (<0.01, 16 d.f.) and for the interaction between S+ vs. S- and proficiency (p<0.05, 16 d.f.) (*Figure 2Hiv*). For a downstream observer focused on the LFP power at the peak of PAC for the rewarded odorant these changes in peak angle variance may reflect a mechanism by which the unrewarded stimulus is devalued while the rewarded odorant is valued. To our knowledge, this is the first

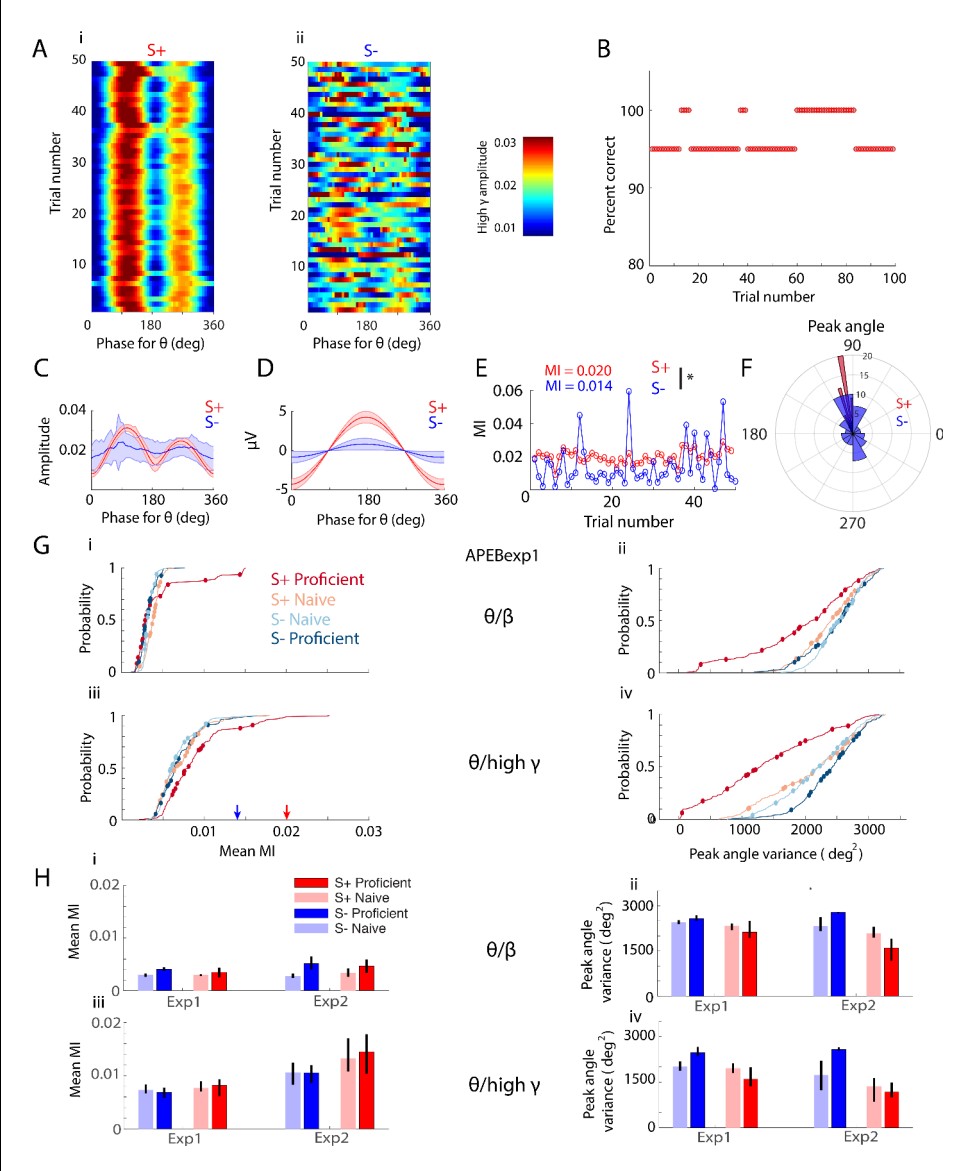

**Figure 2.** The peak angle variance of PAC changes as the animal learns to discriminate odorants. (**A-F**) Example of theta/high gamma phase amplitude coupling for a proficient animal (IAAPexp1 odorant pair). (**A**) Pseudocolor plots showing the phase amplitude relationship for S+ (left) and S- (right) for an example go no-go session. PAC between theta (6–14 Hz) phase and high gamma (65–95 Hz) amplitude was more variable for S- compared to S+. (**B**) Behavioral performance for this session. Over 100 trials, this trained animal discriminated between isoamyl acetate and acetophenone with 95–100% accuracy. A sliding window of 20 trials with a step of one trial was used to calculate percent correct behavioral performance. (**C**) Theta phase distribution for high gamma amplitude (mean ± CI) for S+ (red) and S- (blue). (**D**) Mean theta LFP (mean ±CI) for one cycle for S+ (red) and S- (blue). (**E**) MI per trial for S+ (red) and S- (blue) as a function of trial number. The difference in MI between S+ and S- is statistically significant (ranksum p value < 0.001, n = 50 s- trials, 49 S+ trials). (**F**) Polar histograms for high gamma peak theta angles for S+ (red) and S- (blue). (**G**) Cumulative histograms showing the differences in mean MI (i,iii) and peak angle variance (ii,iv) for theta/beta PAC (i,ii) and theta/high gamma PAC (iii,iv) for the APEB odorant in Exp1 for the S+ and S- odorants for naïve and proficient trial ranges. Peak angle variance for S+ displays a substantial decrease with learning, while mean MI shows small changes. Changes with learning and between events for mean MI and peak angle variance are statistically significant (GLM p value < 0.001, 896 d.f. for theta/ beta and theta/high gamma, 16 electrodes per mouse, 14 mice). Blue and red arrows in Giii are MIs derived from the single-session example in C. (**H**) Bar graph displaying mean MI (i,iii) and peak angle variance (ii,iv) for naïve and proficient trial ranges and S+ and S- events for mean values calculated per experiment for theta/beta (i,ii) and theta/high gamma PAC (iii,iv). GLM yielded significant p values for peak angle variance for theta/high gamma PAC

*Figure 2 continued on next page*

*Figure 2 continued*

for S+ vs. S- (<0.01, 16 d.f.) and for the interaction between S+ vs. S- and proficiency (p<0.05, 16 d.f.). GLM yielded significant p values for peak angle variance for theta/beta PAC for S+ vs. S- and experiment and for the interaction between S+ vs. S- and experiment (<0.01, 16 d.f.). For mean MI theta/high gamma PAC GLM yields significant difference for experiments (p<0.01, 16 d.f.). Finally, there are no significant effects for mean MI for theta/beta PAC (p>0.05, 16 d.f.) and no post hoc tests yield significant differences (p>pFDR).
The online version of this article includes the following figure supplement(s) for figure 2:

**Figure supplement 1.** Example of PAC changes over learning in one session.
**Figure supplement 2.** Modulation index and peak angle variance shown for each odorant pair.

demonstration of dynamic cross frequency interactions changing with learning for early processing in sensory systems.

## Peak phase referenced power increases for S+ and decreases for S- over learning

The existence and evolution of PAC over learning alone do not demonstrate its potential utility to the animal in its decision-making process. In order to assess whether high frequency LFP power at the peak of theta oscillations changes as the animal learns we performed wavelet analysis to extract the power of high gamma (and beta) LFP referenced to specific PAC phases of theta—peak and trough (as defined in *Figure 1E*). We term this analysis phase-referenced (high-frequency) power, or PRP. PRP analysis allows answering the question whether the power of burst high frequency LFP evaluated at a specific phase for PAC changes as the animal learns to differentiate the odorants in the go-no go task.

In *Figure 3A–C*, we show how this PRP analysis was performed. *Figure 3A* shows Morlet wavelet LFP power analysis (*Chery et al., 2014*; *David et al., 2015*) for a single S+ trial characterized by strong, persistent theta power during and after odor delivery (related to respiration and olfactory investigation) accompanied by higher frequency power bursts whose periodicity reflects PAC. *Figure 3B* shows the theta (6–14 Hz)-filtered waveform (*Figure 3Bi*) and the Hilbert-extracted phase (*Figure 3Bii*; *Tort et al., 2010*) co-registered to the one-second epoch extracted from the wavelet LFP power plot shown in *Figure 3Aii*. We proceeded to extract the value of the power (in decibels) referenced to the peak or the trough of PAC measured in S+ trials (peak and trough are defined in *Figure 1E*). The red line in *Figure 3Ci* shows a clear odor-induced increase in the peak-referenced wavelet power for high gamma/theta PAC in this single trial example (*Figure 3Ci*, red line). In contrast, there was little change in power referenced to the trough of PAC (*Figure 3Ci*, blue line). *Figure 3Cii* shows the time course for the average peak (red) and trough (blue) PRP calculated for all trials in this go-no go session for a mouse that was performing proficiently in differentiation of the two odorants.

In order to assess whether PRP changes with learning we performed this analysis for all sessions for naïve and proficient periods. *Figure 3D* shows cumulative histograms for the peak and trough PRP for all mice for the odor period for all go-no go sessions with the IAAPexp2 odorant pair for theta/high gamma PAC. Over the course of learning, peak-related high gamma power increases for S+ and decreases for S- (*Figure 3D* top). In contrast, the changes elicited by learning in trough PRP are smaller (*Figure 3D* bottom). The differences were statistically significant between peak and trough, S+ and S- and naïve and proficient (GLM p values < 0.001, 1016 d.f.). The summary of PRP for peak and trough for theta/high gamma PAC for the odor period are shown for all odorant pairs in *Figure 3E* (peak PRP) and 3F (trough PRP). For all odorant pairs peak PRP increases for S+ over learning and decreases for S-. Trough PRP is weaker yet follows the same trend as the animal learns to discriminate the odorant as peak PRP (GLM p value < 0.001, 6386 d.f. for S+ vs. S-, peak vs. trough, naïve vs. proficient). In addition, the learning-induced changes in PRP are larger for Exp2 (GLM p value < 0.001 for experiment, 6386 d.f.). Similar changes were found for PRP for theta/beta PAC, but there was no difference between peak and trough (not shown).

Finally, we found that theta peak phase-referenced power decreases for S- and increases for S+ regardless of the chemical identity of the odorant. *Figure 4A* shows that after the valence of the odorant is reversed the animal learns to respond to the new rewarded odorant. We asked the

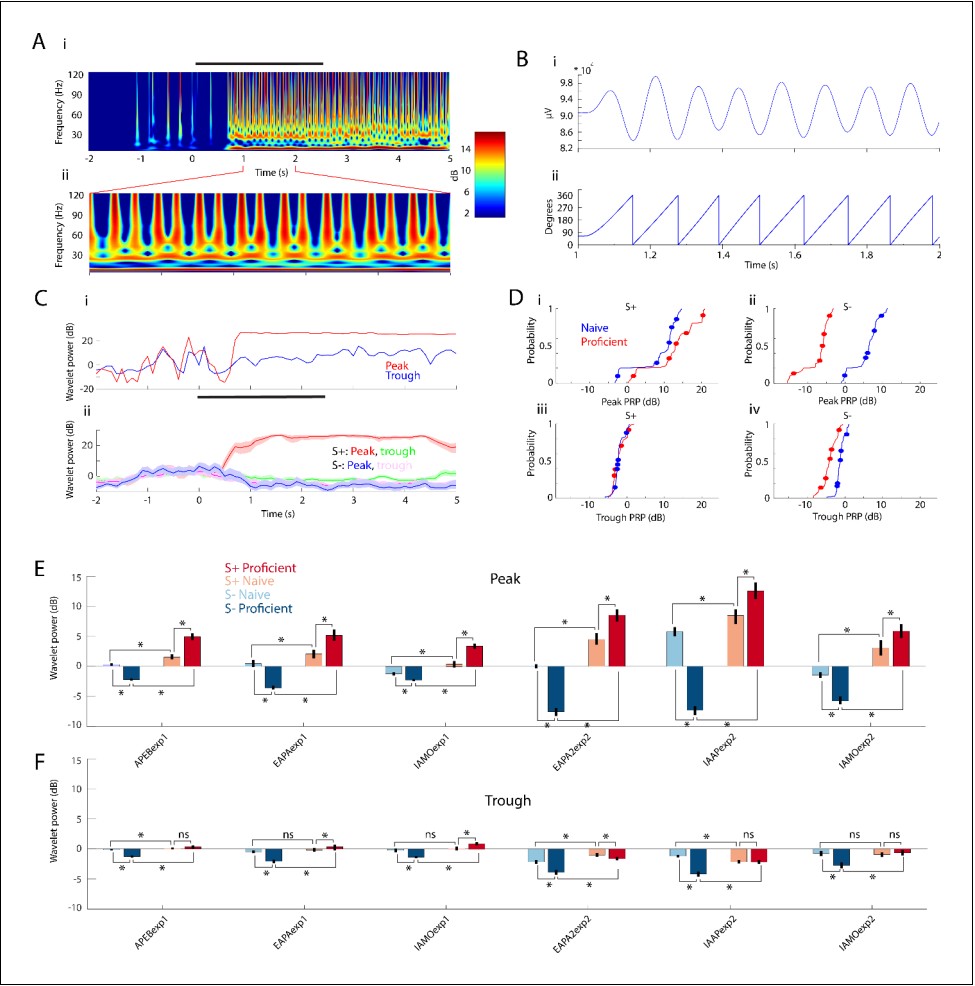

**Figure 3.** PRP increases for S+ and decreases for S- over learning. (A) Example wavelet broadband LFP spectrogram for one S+ trial. Right: Pseudocolor scale for LFP power (in dB). Black bar represents the odor stimulation epoch (2.5 s). i: Full trial length demonstrating pre-odor baseline LFP, odor driven LFP, and reward epoch. ii: One-second epoch during odor stimulation. (B) i: Theta waveform (6–14 Hz) during odor stimulation. ii: Theta phase, extracted with Hilbert transform. (C) i: Single trial wavelet power referenced to either peak (108°, red) or trough (0°, blue) of the high gamma amplitude distribution along the theta phase. ii: PRP (± CI) averaged across the entire session for S+ and S-. Peak PRP for S+ displays the largest odor-induced increase in power that is sustained after the end of odor stimulation. (D) The thin lines are cumulative histograms for PRP for the LFP recorded in each electrode averaged for theta/high gamma for all proficient or naïve trials for mice performing to differentiate odorants in the IAAPexp2 odorant pair. i and ii are peak PRP and iii and iv are trough PRP. i and iii are S+; ii and iv are S-. As the animals learn the task, peak PRP (i and ii) increases for S+ and decreases for S-, whereas trough PRP (iii and iv) is relatively stable. The differences were statistically significant between peak and trough, S+ and S- and naïve and proficient (GLM p values < 0.001, 1016 d.f.). Circles are per mouse averages for PRP. (E–F) Odor-induced changes in high gamma PRP averages for all odorant pairs evaluated per electrode during the odor delivery epoch (0.5–2.5 s). Peak PRP (E) increases for S+ and decreases for S- when the mice become proficient. Trough PRP changes (F) are markedly weaker, yet they follow a similar trend (GLM p value < 0.001, 6386 d.f. for S+ vs. S-, peak vs. trough, naïve vs. proficient). In addition, the learning-induced changes in PRP are larger in Exp2 (GLM p value < 0.001 for experiment, 6386 d.f.).

question whether odorant-induced changes in peak PRP would change when odorant valence was reversed. *Figure 4B and C* show that in proficient mice peak PRP increases when the mouse responds to the rewarded odorant regardless of the chemical identity of the odorant, and PRP decreases when the mouse responds to the unrewarded odorant. GLM analysis yields significant changes for odorant, reversal and the interaction of reversal and odorant (p<0.001, 1051 d.f.). These data show that as the animal learns to differentiate between odorants in the go-no go task there is

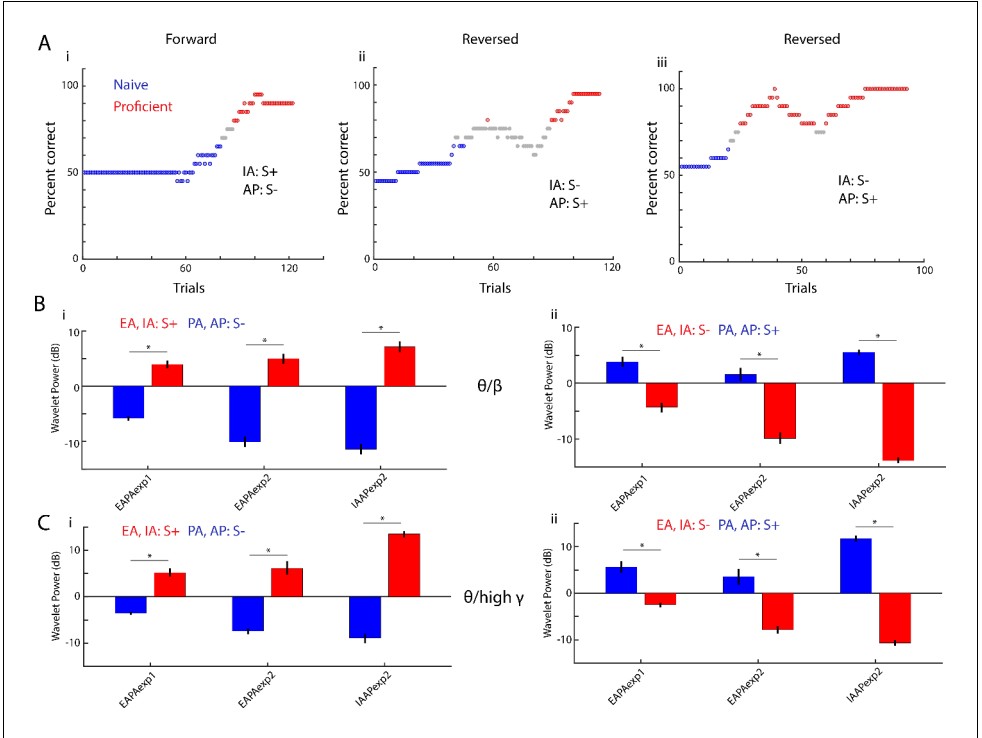

**Figure 4.** Peak theta-phase referenced power switches when odorant valence is reversed. (A) Behavioral performance in three go-no go sessions for a mouse learning to differentiate the IAAPexp1 odorant pair. In the first session the rewarded odorant (S+) was isoamyl acetate (IA) and the unrewarded odorant (S-) was acetophenone (AP). In the other two sessions the valence of the odorant was reversed. Blue:≤65% percent correct performance (naïve) and red:≥80% (proficient). (B–C) Theta/beta (B) and theta/high gamma (C) peak PRP, calculated for trials when the mouse is proficient, switches when odorant valence is reversed. The data are shown for three odor pairs: EAPAexp1 (11 mice), EAPAexp2 (nine mice), IAAPexp2 (three mice). GLM analysis yields significant changes for odorant, reversal and the interaction of reversal and odorant (p<0.001, 1051 d.f.). Asterisks denote post hoc significant differences (p<pFDR = 0.04).

an increase in rewarded odorant-induced peak PRP and a decrease in unrewarded-odorant peak PRP.

## Linear discriminant analysis classifies stimuli using PRP

The finding that LFP power referenced to the peak of the theta oscillation increases for S+ and decreases for S- when the animal learns to differentiate the odors (naïve vs. proficient) raises the question whether a downstream reader evaluating peak power could decode which stimulus is presented. To assess decoding of olfactory stimuli using PRP, a linear discriminant analysis (LDA) was used to set a decision boundary hyper-plane between binary stimulus classes (S+ versus S-) (*Vizcay et al., 2015*) using PRP data referenced to the peak (or the trough) of theta/beta or theta/ high gamma PAC. LDA was trained with PRP from each electrode for each mouse (16 electrodes per mouse) for all trials except one (the training dataset) and then the missing trial (test data) was classified as S+ or S- using the PRP data from that trial. This was performed separately for trials where the mouse was naïve or proficient to odorant discrimination, for each 0.1 s time point throughout the trial and was repeated for all trials. As a control we shuffled the identity of trials in the training set. In addition, we performed a complementary principal component analysis (PCA) of the PRP to visualize the odor-induced divergence in the time course of the first principal component (PC1).

*Figure 5A* shows for theta/beta (i and ii) and theta/high gamma (iii and iv) the time course for PC1 calculated for S+ and S- odorants for naïve (i and iii) and proficient (ii and iv) mice calculated with the PRP for 16 electrodes for one example odorant pair (APEBexp1). In the naïve state PC1 diverged between S+ and S- shortly after water reinforcement for peak and trough theta/beta PRP

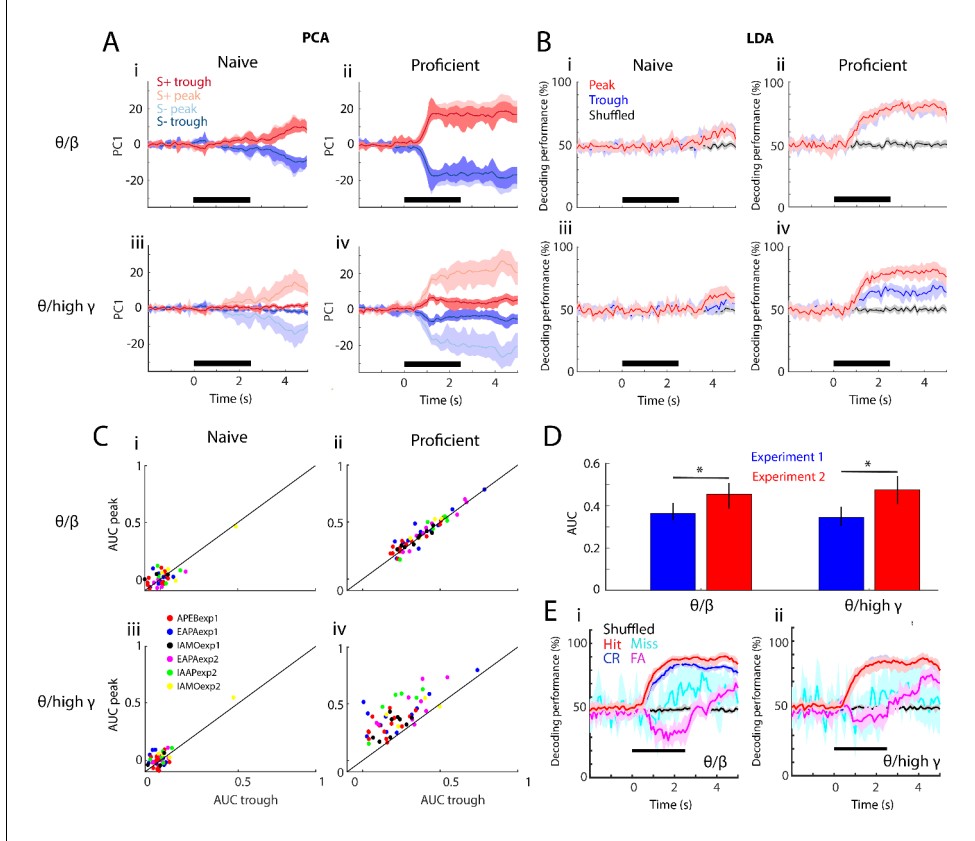

**Figure 5.** Linear discriminant analysis classifies stimuli using peak PRP. (A–B) PCA (in panel A) and linear discriminant analysis (LDA; in panel B) of averaged PRP data for one example odorant pair, APEBexp1. (A) PCA of PRP data from naïve (i,iii) to proficient (ii,iv)—(i,ii): theta/beta, (iii,iv): theta/high gamma. Both PRPs for S+ and S- are discriminable with learning. (B) Decoding performance for the LDA of PRP data over learning (i,iii: naïve, ii,iv: proficient). i,ii: theta/beta, iii,iv: theta/high gamma. In the proficient state, LDA can decode stimulus identity using both PRPs (theta/beta PRP GLM naïve-proficient p value < 0.001, 104 d.f.; theta/high gamma PRP GLM naïve-proficient p value < 0.001, 104 d.f.). Peak and trough carry similar information for theta/beta; both peak- and trough-referenced beta power can be used to decode stimulus identity with learning. In theta/high gamma, peak PRP is significantly better at decoding the stimulus identity than trough PRP (GLM p value < 0.001, 104 d.f.), which is still better than shuffled data. (C) Area under the curve (AUC) for peak versus trough PRP over task learning (i,iii: naïve, ii,iv: proficient) for all odorant pairs and experiments. i,ii: Theta/beta PRP. iii,iv: For theta/high gamma PRP GLM statistics show that peak AUC is significantly higher than trough (p value < 0.001 for six odorant pairs, 184 d.f.) and that there are significant changes over learning (p value < 0.001 for six odorant pairs, 184 d.f.). These data suggest that peak PRP decodes stimulus identity better than trough PRP. (D) Average peak AUC (±CI) for all odorant pairs for theta/beta and theta/high gamma PRP for proficient mice for the two experimental settings. The learning-induced changes in PRP are larger in Exp2 (GLM p value < 0.05 for experiment, 102 d.f., *post hoc p<pFDR = 0.03). (E) Decoding performance for the LDA of PRP data for proficient mice calculated for the different behavioral outcomes of the go-no go task (Hit: red, Miss: cyan, CR: blue, FA: magenta, shuffled: black). i: Theta/beta PRP, ii: Theta/high gamma PRP. A GLM analysis indicates that there are significant differences between all behavioral outcomes except for Hit vs. CR for theta/high gamma (p<0.001 for all outcome pairs, except for Hit vs CR for theta/beta PRP that has p<0.01 and Hit vs. CR for theta/high gamma that has p>0.05, 2110 d.f.). There is no significant difference between experiments (GLM p>0.05, 2110 d.f.). The lines in A,B and E are bounded by the bootstrapped 95% CI..

The online version of this article includes the following figure supplement(s) for figure 5:

**Figure supplement 1.** Simulated LDA performance data illustrating AUC zero and one.

(*Figure 5Ai*) and for peak theta/high gamma PRP (*Figure 5Aiii*, with a smaller change for trough). However, when the animal became proficient PC1 diverged between S+ and S- shortly after odorant addition for theta/beta PRP (*Figure 5Aii*) and theta/high gamma PRP (*Figure 5Aiv*). This analysis suggests that for the proficient animal PRP LFP carries information on the rewarded odorant.

In order to quantify the accuracy for decoding odorant identity using PRP LFP we performed LDA analysis (*Figure 5B*). For naïve animals the LDA could not classify stimulus identity using peak or trough PRP for either theta/beta (*Figure 5Bi*) or theta/high gamma (*Figure 5Biii*). In contrast, when the animal was proficient the precision of the LDA classifier increased shortly after odorant application from chance to values above 70%, significantly higher than the 95% confidence interval of the LDA performed with shuffled trials for both theta/beta (*Figure 5Bii*) and theta/gamma (*Figure 5Biv*) (GLM analysis p value for learning <0.001, 104 d.f.). For theta/beta, both peak and trough PRP LDA performed significantly better than shuffled data for proficient mice (*Figure 5Bii*, GLM p value < 0.001, 104 d.f.). However, for theta/high gamma in proficient animals (*Figure 5Biv*, bottom) LDA discriminated the identity of the stimulus more effectively using peak PRP (GLM p value < 0.001, 104 d.f.).

The difference in LDA performance between peak and trough PRP was further evaluated by calculating the area under the curve (AUC) for LDA performance during the odor application window (0.5–2.5 s after odorant valve opening). AUC is a statistical method to quantify differences between the distributions of values of two variables (*Green and Swets, 1988*). The AUC was normalized and it ranged from zero (random decoding at 50%, *Figure 5—figure supplement 1Ai*) to 1 (100% decoding during the odor period, *Figure 5—figure supplement 1Aii*). AUC is plotted for all odorant pairs for peak and trough PRP in *Figure 5C* for theta/beta (i and ii) and theta/high gamma (iii and iv) for naïve (i,iii) and proficient (ii,iv) conditions. In the proficient state, AUC for peak PRP LDA for high gamma was significantly higher than AUC for trough PRP LDA (*Figure 5Civ*, GLM for AUC peak vs. trough p value < 0.001, 104 d.f.). AUC for beta PRP LDA did not differ between peak and trough (*Figure 5Cii*). Furthermore, *Figure 5D* shows that the AUC was larger for Exp2 (GLM p value < 0.001 for experiment, 6386 d.f.).

Finally, we asked the question whether decoding performance differs when mice make mistakes in the go-no go task. If PRP encodes for the chemical identity of the odorant we would expect that decoding performance would be similar between Hit/Miss and CR/FA during the first second of the trial when the animal keeps their head in the odor port. If PRP encodes for contextual odorant identity decoding performance would differ between correct responses (Hit and CR) and mistakes (Miss, FA) shortly after the animal detects the odorant. *Figure 5E* shows the results of the per trial analysis of decoding performance for LDA with theta/beta PRP (i) and theta/high gamma PRP (ii) sorted by behavioral outcome. Interestingly, shortly after addition of the odorant there is a divergence in decoding performance between trials with correct decision (Hit, CR) and mistakes (Miss, FA). Shortly after addition of the odorant decoding performance for FA drops below 50% (it tends to encode for the opposite odor), and for Miss trials the decoder performs like the shuffled trials. Decoder performance increases above 50% for both FA and Miss towards the end of the trial. Taken together with the shift of PRP in the reversal experiments (*Figure 4*) the difference in LDA encoding between correct trials and mistakes is further evidence that PRP encodes for the contextual identity of the odorant (as opposed to the chemical identity). A GLM analysis indicates that there are significant differences between all behavioral outcomes except for Hit vs. CR for theta/high gamma (p<0.001 for all outcome pairs, except for Hit vs CR for theta/beta PRP that has p<0.01 and Hit vs. CR for theta/high gamma that has p>0.05, 2110 d.f.). There is no significant difference between experiments (GLM p>0.05, 2110 d.f.).

To our knowledge, this is the first demonstration of high gamma phase-referenced LFP power developing differential information carrying capacity in early sensory systems with learning.

## Decision-making takes place at the same time for peak PRP LDA and licks

After learning that using peak PRP LDA can decode stimulus identity, we wondered how decision-making times estimated based on either the behavioral response of the animal in the go-no go task (licks) or classification of the stimulus by PRP LDA relate to each other. In addition, we asked whether the decision-making time calculated with peak PRP LDA differed from decision-making time estimated using trough PRP LDA. Decision-making time was defined as the time when there is a

statistically significant difference between S+ and S- in odorant classification (estimated separately for licks and LDA).

*Figure 6A–B* show examples of licks by a mouse in the go-no go task for the EAPAexp2 odorant pair. *Figure 6A* shows the licks detected at the waterspout for 25 S+ trials when the animal was performing proficiently while *Figure 6B* shows the lick traces for 25 S- trials in the same session. As in previous studies (*Doucette and Restrepo, 2008*), we estimated the p value for lick decision-making by computing in each 0.1 s time bin the ranksum test for the difference between licks (scored 0 for no lick vs. one for lick). An example of the time course of the p value for licks for mice performing the go-no go task for EAPAexp2 is shown by the green line in *Figure 6Ci*. The p value drops below 0.05 ~ 250 msec after odorant addition. We defined the lick decision-making time as the time point when the p value for difference in licks between S+ and S- falls and stays below 0.05 after odor application.

LDA decision-making time was estimated in a similar manner by computing the p value for a ranksum test for the difference between predictions (scored one for correct prediction vs. 0 for incorrect prediction) for PRP LDA decisions evaluated after pooling data for all mice for each odor pair. As explained under supplemental information (*Figure 6—figure supplement 1*) pooled animal data were used as opposed to per animal PRP LDA because this resulted in consistent estimation of earlier decision-making times. Examples of the time courses of the p values for PRP LDA performing the go-no go task for EAPAexp2 are displayed in *Figure 6Ci* for theta/high gamma PAC (red line: peak, blue line: trough). Before odorant presentation the p value fluctuates and sometimes falls

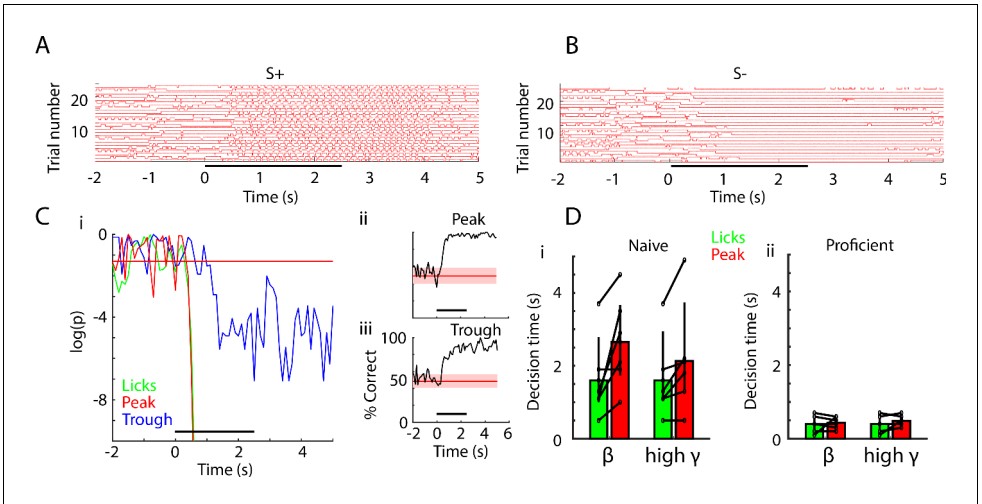

**Figure 6.** Decision-making time does not differ between licks and peak PRP LDA decoding of odorant identity. (**A**) Lick time course for 25 S+ trials in one session of EAPAexp2 for one proficient mouse. Horizontal black bar indicates odor delivery epoch for all time course subpanels. (**B**) Lick time course for 25 S- trials in the same session as in A. Mouse refrains from licking in the presence of the unrewarded odorant. (**C**) i: p value time courses for difference between S+ and S- for licks and peak PRP LDA predictions. ii and iii: decoding performance for PRP LDA. Decision making times for both licks and PRP LDA were calculated using data pooled for all mice performing odorant discrimination with the EAPAexp2 odorant pair. i: p value for the ranksum test compares the time-course for decision-making using licks for S+ vs. S- (green) or LDA prediction of odorant identity with peak (red) or trough (blue) PRP. This example illustrates that that peak PRP LDA prediction and lick divergence perform similarly, while LDA prediction with trough PRP is significantly slower in stimulus decoding. Horizontal red line indicates p=0.05. ii and iii: LDA decoding performance appears to be faster and more accurate using peak PRP (ii) vs. trough PRP (iii). (**D**) Decision times for licks and for pooled animal peak theta/beta and theta/high gamma PRP LDA in naïve (i) and proficient (ii) mice. Decision times were significantly reduced with task learning (GLM p value < 0.001, 44 d.f.). Decision times for lick vs. PRP LDA and theta/beta vs. theta/high gamma did not differ (GLM p values > 0.05, 44 d. f.).

The online version of this article includes the following figure supplement(s) for figure 6:

**Figure supplement 1.** Decision times are faster for peak PRP LFP LDA computed when data are pooled across mice.

below 0.05 transiently, while after odor presentation the p values for peak PRP fall below 0.05 sharply and continuously at approximately the same time as lick p value (~250 msec). Trough PRP p value for LDA decoding performance drops below 0.05 later (~950 msec) and the p values are higher than peak p values (compare red and blue lines in *Figure 6Ci*). Thus, in this example lick decision-making time is similar for licks and peak PRP LDA, and is slower for trough PRP LDA.

We proceeded to calculate decision-making times for licks and peak PRP LDA for all odorant pairs. Decision-making times for LDA decoding performance via peak PRP and lick for all odorant pairs for pooled mouse data are shown in *Figure 6D* for naïve (*Figure 6Di*) an proficient (*Figure 6Dii*) mice. Decision times were reduced significantly with task learning (GLM p value < 0.001, 44 d.f.). However, decision-making times did not differ between lick and peak PRP LDA (GLM p value > 0.05, 44 d.f.).

## Dimensionality of PRP LFP space decreases when the animal learns to discriminate the odorants

*Figure 7A* shows the change in the scatter pattern of the first two principal components (PCs) of the theta/high gamma peak PRP calculated for pooled data from all mice as the animals learn to differentiate the two odorants for the EAPAexp1 odorant pair. *Figure 7Ai* shows that when the mice are naïve S+ (red) and S- (blue) points are mixed in the 2D PC space before and during odorant application. In contrast, *Figure 7Aii* shows that the 2D PC patterns become clearly separable when the animals are proficient. There appears to be a change in the geometrical arrangement of the points in principal component space suggesting that there is a change in dimensionality as the animal learns to differentiate the odorants (*Litwin-Kumar et al., 2017*).

In order to provide a more nuanced analysis of whether the number of independent signals in the PRP LFP data measured by the 16 electrodes changes as the animals learn to discriminate between odorants we calculated a quantitative measure that characterizes dimensionality in the PRP LFP space (see Materials and methods). The 16 electrodes are closely spaced (12.5 µm diameter electrodes in the same tetrode are directly abutted against each other and tetrodes are separated by 100–200 µm in both preparations: Exp1 with optetrodes and Exp2 with tetrodes). Because of this close proximity the extracellular field potentials would not be expected to differ greatly (*Gold et al., 2006*) and therefore the dimensionality of the PRP LFP space would be expected to be close to one. Interestingly, when the dimensionality was calculated on a per mouse basis where the maximum dimensionality (M) is 16 we found that the average dimensionality for theta/high gamma PRP LFP for electrodes in Exp2 was indeed close to one, but it was significantly higher (2-4) in Exp1, with the highest dimensionality for PRP LFP trough (*Figure 7—figure supplement 1Ai*, GLM p value < 0.001 for peak vs. trough and Exp1 vs. Exp2, 379 d.f.). We found similar differences in dimensionality between Exp1 and Exp2 for theta/beta PRP LFP, but in this case there was no difference between peak and trough dimensionality (*Figure 7—figure supplement 1Aii*, GLM p value < 0.001 for Exp1 vs. Exp2 and >0.05 for peak vs. trough, 379 d.f.). This indicates that even when measured using electrodes located within 100–200 µm the LFP carries information on multiple independent components.

Interestingly, the dimensionality for peak PRP LFP for theta/high gamma calculated on a per mouse basis decreased after addition of the odorant for both experiments and the decrease in dimensionality was larger for proficient mice. Examples of the dimensionality time course for peak PRP LFP for theta/high gamma calculated on a per mouse basis are shown for two odorant pairs in *Figure 7—figure supplement 1Bi* (Exp1) and *Figure 7—figure supplement 1Bii* (Exp2). A GLM analysis of the dimensionality for peak PRP LFP calculated on a per mouse basis for all odorant pairs indicates that there are significant differences in theta/high gamma peak PRP LFP dimensionality for proficient vs. naïve, time course, experiments, and for interactions of these factors (except for the interaction of experiments x proficient vs. naïve) (GLM p values < 0.001, 6098 d.f.). For theta/beta peak PRP LFP dimensionality we found similar results with the exception that there was no difference between peak and trough (*Figure 7—figure supplement 1Ci and Cii*, GLM p values < 0.001, 6098 d.f.).

Finally, when the dimensionality for peak PRP LFP was calculated for LFPs recorded from all mice the difference in the decrease in dimensionality between naïve and proficient became more evident. Time courses for pooled mouse peak PRP LFP dimensionality for theta/high gamma are shown in *Figure 7B* for Exp1 (7Bi) and Exp2 (7Bii). Peak theta/high gamma PRP LFP dimensionality decreases markedly when the odorant is delivered for both Exp1 and Exp2 for proficient mice and this

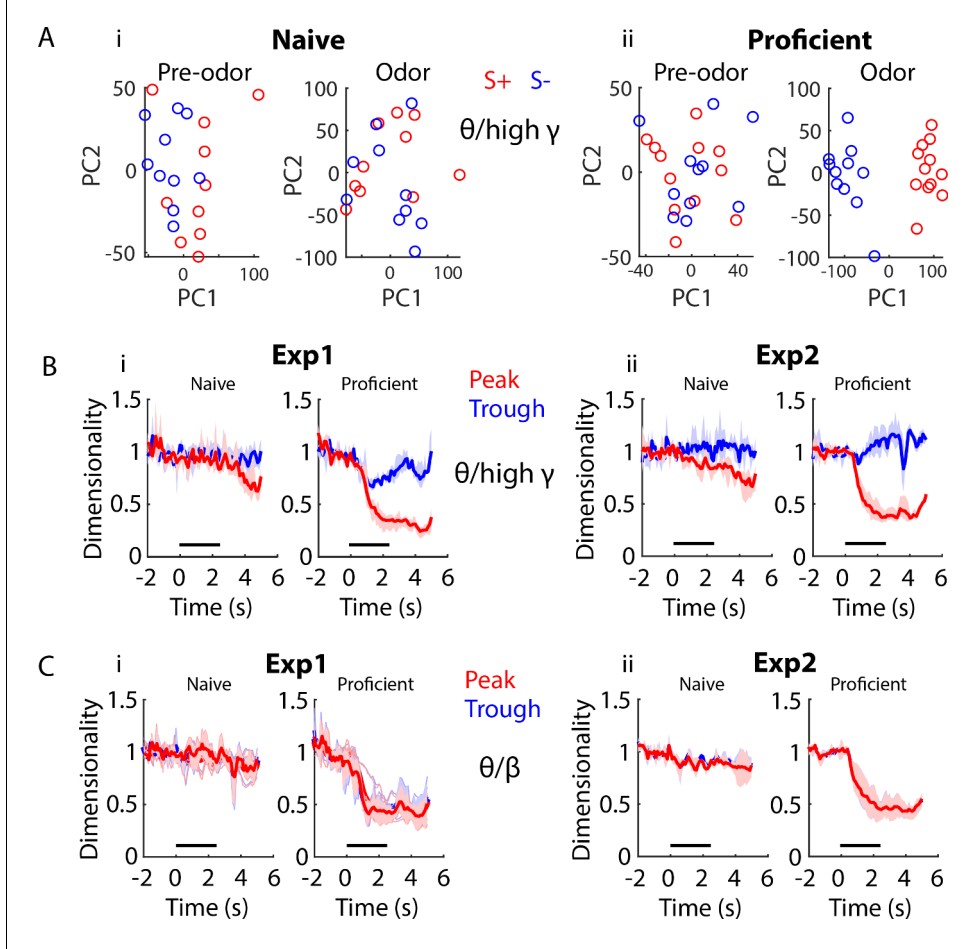

**Figure 7.** Dimensionality changes with learning and during the time course of the trial. (A) Scatter plot showing the first two principal components for twenty trials of a PCA for peak theta/high gamma PRP recorded in mice learning to differentiate the odorants in the EAPAexp1 odorant pair. The input to the PCA was the peak PRP recorded in 16 electrodes pooled for all mice. Principal components are intermingled between S+ (red) and S- (blue) trials in the naïve state (left) in pre-odor and odor-delivery epochs, yet they separate during odor delivery upon task acquisition (right). i. Naïve animals. ii. Proficient animals. (B) Within-trial peak (red) and trough (blue) theta/high gamma PRP dimensionality time course for data pooled across mice is shown for Exp1 (i) and Exp2 (ii), normalized to pre-trial dimensionality. A GLM analysis indicates that there are significant differences for time course, proficient vs. naïve, experiments and peak vs. trough, and for pair interactions of these factors (GLM p values < 0.001, 1688 d.f.). (C) Within-trial peak (red) and trough (blue) theta/beta PRP dimensionality time course for data pooled across mice is shown for Exp1 (i) and Exp2 (ii), normalized to pre-trial dimensionality. GLM analysis found statistically significant differences for time course, and proficient vs. naïve, and for interaction of these factors (GLM p values < 0.001, 1688 d.f.), and there was a statistically significant difference between experiments (GLM p value < 0.01, 1688 d.f.). However, there was no difference between peak and trough (GLM p value > 0.05, 1688 d.f).

The online version of this article includes the following figure supplement(s) for figure 7:

**Figure supplement 1.** PRP dimensionality estimated per mouse.

decrease is larger for proficient compared to naïve mice (compare naïve vs. proficient in *Figure 7Bi and ii*). A GLM analysis indicates that there are significant differences for time course, proficient vs. naïve, experiments and peak vs. trough, and for interactions of these factors (GLM p values < 0.001, 1688 d.f.). For theta/beta peak PRP LFP dimensionality for pooled mice we found similar results (*Figure 7Ci and 7Cii*), but there was no difference between peak and trough. GLM analysis found statistically significant differences for time course, and proficient vs. naïve, and for interaction of these factors (GLM p values < 0.001, 1688 d.f.), and a statistically significant difference

between experiments (GLM p value < 0.01, 1688 d.f.). However, was no difference between peak and trough (GLM p value > 0.05, 1688 d.f). Therefore, for both the per animal and the pooled animal PRP LFP space there is a decrease in dimensionality as the animal learns to discriminate between odorants, but this difference is more evident for the pooled animal data. This is evidence for a learned tightening of dimensionality for the odor representation manifold (defined as the independent neural components representing the odor).

## Discussion

We asked whether information carried by high frequency oscillations in brain regions involved in early sensory processing changes when observed at different phases of theta frequency oscillations. Specifically, we asked whether information on contextual odorant identity conveyed by LFP power of high gamma and beta frequency oscillations of the OB differs when the downstream observer filters the input through different theta LFP phase windows. We focused on the OB where it has been suggested that amplitude modulation may be used by higher-order processing centers such as PC to decode stimulus identity (*Freeman and Schneider, 1982*; *Heck et al., 2019*). We found that identity could not be decoded from beta or high gamma OB LFP power when the mouse was naïve to the identity of the reinforced odorant (left panels in *Figure 5B*). However, the contextual identity of the odorant could be decoded from the PRP after the animal learned to discriminate between the odorants (right panels in *Figure 5B*). Furthermore, decoding was more accurate when the high gamma LFP power was observed at the peak of the theta oscillation, but it was similar at both the peak and trough phases for beta LFP oscillations. LDA correctly identified the stimulus with a time course for decision-making similar to the behavioral readout (licking on the water-delivery spout, *Figure 6*). Thus, stimulus representation by theta phase-referenced beta and high gamma oscillations of the OB evolves over the course of learning. Importantly, our findings were replicated in two different experiments (Exp1 and Exp2) differing in genotype, electrode location, device implanted (tetrode or electrode) and odorant pairs used (see *Supplementary file 1*-Table S1). We did find differences in measured parameters between experiments, and these are likely due to methodological differences between Exp1 and Exp2 such as electrode location and genotype. Regardless of the reason for differences between experiments it is important for this study that the findings on PRP are found for both experiments under substantially different experimental conditions. To our knowledge this is the first demonstration that decoding of stimulus identity from theta phase-referenced high gamma oscillations, reflecting synchronized neuronal firing, changes with learning in brain areas involved in early sensory processing.

The importance of PAC in information processing is based on the notion that different information is carried at different phases of the slow carrier oscillation. This discretization of information is a potential mechanism for encoding short-term memories (*Lisman and Idiart, 1995*) and more broadly as a neural communication protocol, the theta-gamma neural code (*Hopfield, 1995*; *Lisman and Jensen, 2013*). Circuit activation at theta frequency was discovered to be essential for long-term potentiation in the rabbit hippocampus (*Bliss and Lomo, 1973*), arguing that neural information transfer may go beyond mean rate codes and involve spike timing with respect to oscillatory timekeepers. Subsequent studies indicated that temporal windows set by theta cycles allow for local circuit interactions and a considerable degree of computational independence in subdivisions of the entorhinal cortex-hippocampal loop (*Mizuseki et al., 2009*). For CA1, it was hypothesized that theta partitions the encoding of new information versus the retrieval of stored information (*Hasselmo et al., 2002*). Indeed, optogenetic recruitment of fast spiking inhibition in CA1 delivered at specific theta phases altered murine performance in a spatial navigation task (*Siegle and Wilson, 2014*). When the mice were encoding a location, triggering of inhibition of CA1 at theta peaks improved performance, whereas inhibition at theta troughs improved performance during the retrieval phase. Recently, age related cognitive decline in working memory in humans was reversed temporarily via transcranial alternating-current stimulation frequency-tuned to the endogenous theta peak frequency of each individual's frontotemporal network (*Reinhart and Nguyen, 2019*). It appears that working memory is undergirded by theta-gamma PAC and that inter-regional communication is facilitated by theta phase coherence (*Daume et al., 2017*; *Reinhart and Nguyen, 2019*). Thus, the theta LFP which has been cited as one of the most global oscillations in the brain acts as a

timekeeper (**Siegle and Wilson, 2014**) that is coherent across numerous cortical and subcortical structures arguing for its role in transfer of discrete chunks of information (**Buzsáki, 2002**).

Here we asked whether odor identity information was carried at different phases of theta. Our study of PAC in the OB involved learning of odor discrimination in the go no-go task. In orbitofrontal cortex, Van Wingerden et al. (**van Wingerden et al., 2014**) showed that this task elicited strongest theta/gamma PAC during odor sampling preceding correct decision-making, which they attributed to temporal coordination between cell assemblies responsible for stimulus representation and reward association. The OB circuit is characterized by beta and gamma oscillations nested in a slower theta carrier wave pertaining to respiration (**Buonviso et al., 2003**; **Heck et al., 2019**; **Kay, 2005**; **Kepecs et al., 2006**; **Rebello et al., 2014**; **Rojas-Líbano et al., 2014**). However, Granger directionality analysis shows that the theta oscillation in the OB is also influenced by hippocampal slow oscillations indicating that the origin of OB slow carrier oscillations is complex, involving both respiration and centrifugal input from downstream brain regions whose relative contribution likely changes under different behavioral states (**Nguyen Chi et al., 2016**). Cyclical activation of OSNs through respiration (**Adrian, 1950**; **Bressler, 1988**; **Rosero and Aylwin, 2011**; **Zhuang et al., 2019**) provides input to mitral/tufted (M/T) cells, and coordinated activity by M/T cells in the OB caused by mitral cell-granule cell reciprocal dendrodendritic interactions establishes gamma oscillations (**Arnson and Strowbridge, 2017**; **Osinski and Kay, 2016**; **Pouille and Schoppa, 2018**; **Schoppa, 2006**; **Stopfer et al., 1997**) (see review by **Heck et al., 2019**). Gamma LFP is thought to represent the stimulus identity (**Bathellier et al., 2006**; **Beshel et al., 2007**; **Li and Cleland, 2017**; **Rojas-Líbano and Kay, 2008**) and beta LFP oscillations (15–35 Hz) have been linked to learned reward (**Martin et al., 2006**). However, the classification of these two bandwidths as carrying learning vs. odor identity information is not always appropriate, and **Frederick et al. (2016)** proposed that they represent different cognitive states that vary depending on particular behavioral demands. PAC is present in the OB (**Buonviso et al., 2003**; **Rojas-Líbano et al., 2014**) and it may be of great albeit relatively unexplored utility. We contributed to the understanding of which information is carried by different frequency oscillations by asking whether the high gamma and beta oscillations carry different information when they are observed within particular phase windows of the theta carrier wave (**Figure 5**). We hypothesized that PAC may be used to convey olfactory stimulus information to theta phase-synchronized downstream regions like PC (**Kepecs et al., 2006**; **Lisman and Buzsáki, 2008**).

We demonstrate that PAC changes in the OB during associative learning (**Figure 2**). Peak angle variance increased for the unrewarded odorants with learning. The theta phase-referenced PRP of the high gamma and beta LFPs also follows this trend (**Figure 3**), increasing for S+ and decreasing with S- over learning. The changes in PRP correlating with learning may reflect contrast enhancement that facilitates the discrimination between similar stimuli by a downstream observer. Neuronal adaptation to facilitate contrast enhancement has been shown at the level of the OSNs (**Haney et al., 2018**), the glomerular layer (**Cavarretta et al., 2016**), and the granule cell layer (**Adams et al., 2019**) of the OB. We speculate that the concurrent increase in PRP for S+ and decrease in PRP for S- reflects adaptational mechanisms in the OB to enhance contrast between similar stimuli with different associated values. Finally, given that there is coordination of oromotor actions through brainstem circuits (**Moore et al., 2014**), and that studies have shown development of coordinated sniffing and licking as rodents learn to differentiate odorants (**Jordan et al., 2018**; **Lefèvre et al., 2016**; **Rosero and Aylwin, 2011**) it is likely that the changes found in PRP as the animals learn to differentiate the odorants reflect an interaction with changes in licking and sniffing.

To quantify whether these changes in PAC increased discriminability between odorants using PRP we asked whether the stimulus that was delivered could be decoded from PRP data. Previous investigations have decoded olfactory stimuli from spike counts aligned to sniffing in the mouse (**Cury and Uchida, 2010**; **Gschwend et al., 2012**), in the antennal lobe of the locust (**Stopfer and Laurent, 1999**), and M/T neural activity assayed through calcium transients in mice (**Chu et al., 2016**), but trial classification via PAC remains unexplored. Here, we performed LDA on PRP data to determine whether a single hyperplane could differentiate between stimuli as the animals learned the task. **Figure 5B** indicates that the identity of the stimuli cannot be decoded from PRP recorded in the naïve mice, but that the stimuli can be decoded when the mice learn to discriminate the odorants. Furthermore, decoding the stimuli from peak high gamma PRP is significantly better than trough PRP. Thus, decoding of the odorant identity by downstream observer using a phase window

for the slow carrier oscillation could detect the identity of the odorants after the animal learns to differentiate between odors in the go no-go task. Finally, this study contributes to the mounting evidence that the OB is an important locus for learning-related activity changes (*Abraham et al., 2014*; *Doucette et al., 2011*; *Doucette and Restrepo, 2008*; *Koldaeva et al., 2019*; *Li et al., 2015*).

We wondered how timing for decision-making would relate comparing stimulus decoding with PRP with decision-making estimated from the behavioral readout of the olfactory-driven decision in the go no-go task (differential licking between the two odorants). The time course for the behavioral decision, reflected in the graph of p value for differential licks, is overlaid by the p value time course for peak PRP decoding (*Figure 6C*, right) with trough PRP lagging behind. Neither beta nor high gamma peak PRP p values are discriminable from lick p values in terms of decision time in this task. As shown in *Figure 6D*, for proficient animals licking differs between odorants within about 0.5 s from stimulus onset, at the same time that PRP decoding becomes significant. This is slower than discrimination times for odorant responses recorded in mice performing two alternative forced choice odorant discrimination or receiving odors passively where discrimination takes place within the first respiratory cycle (~250 ms) after stimulus onset (*Cury and Uchida, 2010*; *Short and Wachowiak, 2019*; *Shusterman et al., 2011*). However, the 0.5 s decision-making time is consistent with studies with the go no-go task where there is not a strong motivation for the animal to stop licking for the unrewarded odorant (*Abraham et al., 2004*; *Doucette and Restrepo, 2008*; *Jordan et al., 2018*; *Rinberg, 2006*)—with the exception of rapid detection of optogenetic activation of glomerular input in the go no-go task (*Smear et al., 2011*). Thus, we conclude that PRP is an accurate reflection of decision-making in olfactory discrimination.

Interestingly, in contrast with theta/high gamma PRP where the decoding performance for peak and through differ substantially, for theta/beta PRP there is no difference in decoding performance between peak and trough (compare *Figure 5Bii and 5Biv*). Increases in OB beta LFP for rodents engaged in odorant discrimination tasks take place after the onset of changes in gamma LFP and it has been postulated that beta LFP represents systemwide coherent states perhaps conveying centrifugal modulation from downstream brain areas to the OB (*David et al., 2015*; *Frederick et al., 2016*; *Kay, 2014*). If beta oscillations indeed represent centrifugal feedback our findings suggest that centrifugal modulation does not differ through phases of the theta oscillation and would therefore excerpt modulation on high frequency circuit activity occurring at different theta phases. Here we show that peak-referenced high gamma oscillation power encodes for the contextual identity of the odorant (is the odorant rewarded?). What information could be conveyed at other theta phases is an open question. Perhaps asynchronous M/T firing carries chemical odorant information in the theta trough.

Finally, we turn to dimensionality. The measurements in the experiments presented here are inherently low dimensional because: 1) The animal is discriminating between two odors, and the behavior is constrained by the go no-go task (*Gao et al., 2017*) and 2) While the number of independent degrees of freedom for inputs to the OB is potentially large (~2000 glomeruli), the four tetrodes are close together and they should sample a small subset of these input dimensions. Indeed, because the tetrodes are within 200 µm of each other (and electrodes are adjacent within tetrodes), the LFP recorded from each electrode is likely to be similar, assuming that the cellular milieu around the electrodes does not separate the inputs (*Gold et al., 2006*). Interestingly, the dimensionality measured from the 16 electrodes (four tetrodes) varies between 1 and 6 suggesting that subsets of electrodes are detecting different signals (*Figure 7—figure supplement 1A*). However, when the mice learn to discriminate the dimensionality decreases in the presence of the relevant odorant (*Figure 7* and *Figure 7—figure supplement 1B and C*), suggesting that population representation is a modifiable aspect of OB processing. This likely reflects experience-dependent, attention (*Canas and Jones, 2010*; *Niv et al., 2015*). In the naïve state, relevant (rewarded) dimensions are unknown to the mouse, and a downstream structure sampling from the recorded population would receive ambiguous signaling in the presence of the rewarded odorant without decoding or reducing the input information. We find a large decrease in dimensionality post-learning, suggesting that OB now enhances a relevant and invariant signal to allow for unambiguous odor detection. Attention, guided by reward prediction error, likely biases/constrains the learning to the relevant dimensions (*Leong et al., 2017*)—in this task, odor identity.

The changes in signal processing elicited by learning, habituation, and passive odorant exposure in the OB, the earliest processing center in the olfactory system, are remarkable. Why is processing

in this circuit so plastic and influenced by centrifugal input from downstream targets, such as PC, and from a variety of neuromodulatory centers (*Gire et al., 2013*; *Linster and Cleland, 2016*)? Likely the reason is that this processing center receives massive parallel glomerular input and projects M/T output of high dimensionality to downstream brain processing areas (*Chae et al., 2019*). Furthermore, unsupervised learning modifying cellular interactions in the OB would increase dimensionality (*Hiratani and Latham, 2019*). Therefore, odors must be identified in the midst of a high dimensionality cocktail party (*Rokni et al., 2014*). Based on the present data showing a large increase in the accuracy for decoding of contextual odor identity from the PRP as the animal learns to differentiate the odorants we speculate that when a subset of the degrees of freedom of the sensory input become behaviorally relevant, centrifugal feedback elicits supervised learning that results in a decrease in dimensionality of the output from the OB to downstream brain areas. Thus, the fact that peak theta-referenced high gamma power encodes information on whether the odorant is rewarded may simply reflect the fact that the transfer of sensory information on the rewarded odorant is being prioritized in the OB circuit. This would result in more accurate discrimination of the input channels that are relevant to the current task. Whether (and how) supervised learning-induced OB circuit plasticity results in a decrease in dimensionality of the output remains to be determined.

# Materials and methods

**Key resources table**

| Reagent type | Reagent or resource | Source | Identifier | Additional information |
|---|---|---|---|---|
| Chemical compound, drug | Isoamyl acetate | Sigma-Aldrich | Cat#123-92-2 | Odorant |
| Chemical compound, drug | Ethyl acetate | Sigma-Aldrich | Cat# 141-78-6 | Odorant |
| Chemical compound, drug | Propyl acetate | Sigma-Aldrich | Cat# 109-60-4 | Odorant |
| Chemical compound, drug | Mineral oil | Sigma-Aldrich | Cat# 8042-47-5 | Odorant |
| Chemical compound, drug | Acetophenone | Sigma-Aldrich | Cat# 98-86-2 | Odorant |
| Chemical compound, drug | Ethyl benzoate | Sigma-Aldrich | Cat# 93-89-0 | Odorant |
| Other | Nickel-chrome wire | Sandvik | Cat#PX000002 | Tetrode fabrication |
| Other | 16 Channel Electrode Interface Board | Neuralynx | EIB-16 | Tetrode fabrication |
| Other | LFP Data | GigaDB | http://doi.org/ 10.5524/100699 | *Losacco et al., 2020* |
| Software, algorithm | Analysis code | This paper | Available on github and by request | https://github.com/ restrepd/drgMaster |
| Strain, strain background | Mouse: C57BL/6J | Jackson Lab | RRID: IMSR_JAX:000664 | Male mice |
| Strain, strain background | Mouse: Tg(Dbh-cre) KH212Gsat/ Mmucd | Mutant Mouse Resource and Research Center | 032081-UCD, RRID:MMRRC | All sexes used |

*Continued on next page*

*Continued*

| Reagent type | Reagent or resource | Source | Identifier | Additional information |
|---|---|---|---|---|
| Strain, strain background | Mouse: eNpHR3.0, 129S-Gt(ROSA) 26Sortm39(CAG-hop/EYFP) Hze/J | Jackson Lab | 014539, RRID: SCR_004633 | All sexes used |
| Software, algorithm | MATLAB_R2018a | Mathworks | RRID: SCR_001622 | |
| Software, algorithm | Illustrator | Adobe | RRID: SCR_010279 | |
| Software, algorithm | Photoshop | Adobe | RRID: SCR_014199 | |
| Strain, strain background | Mouse: OMP-hChR2V | In house | OMP-hChR2V | *Li et al., 2014* |

## Animals

The study was performed in two sets of experiments. For the first set of experiments (Exp1), DBH-eNpHR3.0 mice were used as per *Ramirez-Gordillo et al. (2018)*. These mice express halorhodopsin in cells expressing Cre under the *Dbh* promoter (DBH-Cre eNpHR3.0) in LC-NA neurons and were produced by crossing DBH-Cre mice (032081-UCD, Mutant Mouse Resource and Research Center) with mice expressing halorhodopsin in a Cre-dependent fashion [eNpHR3.0, 129S Gt(ROSA) 26Sortm39(CAG-hop/EYFP) Hze/J, Jackson labs]. OMP-hChR2V (*Li et al., 2014*) or C57BL/6 mice (Jackson stock number: 000664) were used for the second set of experiments (Exp2). All animal procedures were performed under approval from the Institutional Animal Care and Use Committee (IACUC) at the University of Colorado Anschutz Medical Campus under guidelines from the National Institutes of Health. The number of mice used in each experiment is shown in *Supplementary file 1*-Table S1.

## (Op)tetrode implantation

Surgery was performed under approval from the Institutional Animal Care and Use Committee (IACUC) at the University of Colorado Anschutz Medical Campus, using aseptic technique. As per *Li et al. (2015)*, tetrode boards (EIB-16, Neuralynx) with optional fiber optic cannula for photostimulation (optetrodes) were populated with four tetrodes consisting of four 12.5 µm nichrome wires coated with polyimide (Sandvik RO800). For optetrode fabrication the tetrode boards received the addition of a fiber optic cannula 105 µm in diameter (Thorlabs FG105UCA) with 1.25 mm OD ceramic ferrule (Precision Fiber Products) for coupling to a DPSS laser (473 nm, Shanghai Laser and Optics Century). The optogenetic experiments of *Ramirez-Gordillo et al. (2018)* are not included in this study. In the rest of the text we refer to the inserts as (op)tetrode with the understanding that optetrodes were implanted for Exp1 while optetrodes or tetrodes were implanted for Exp2. Electrode tips were electroplated to 0.2–0.4 MΩ impedance.

Two-month-old male mice were anesthetized with 5% isoflurane in oxygen. Intraperitoneal ketamine/xylazine (100 mg/kg and 10 mg/kg respectively) was then administered along with 100 µl of 2% lidocaine injected subcutaneously over the skull. After the mouse was found to be unresponsive to a toe pinch, the animal's head was then secured in the stereotaxic apparatus (Narishige SR-5M-HT) and the skull was leveled ($\leq$50 µm difference DV between bregma and lambda). Gentamycin ophthalmic ointment was applied to the eyes to maintain hydration. After incising the skin overlaying the skull, the periosteum was cleared with 15% $H_2O_2$. A manipulator (Sutter MP-285) was zeroed at bregma and midline and the target location for OB implantation was marked with respect to bregma (Exp1: AP +4.28 mm, ML +0.4 mm, Exp2: AP +4.28 mm, ML +0.5 mm).

A craniotomy performed at this site (Marathon III drill) exposed dura mater which was removed prior to implantation. Another craniotomy was performed more caudally for implantation of one ground screw (Plastics1 00–96 × 1/16). The (op)tetrode was positioned above the craniotomy over the OB while the ground wire was wrapped around the ground screw with the connection coated in silver paint (SPI Flash-Dry silver conductive paint). After securing the ground screw to the skull, the (op)tetrodes were lowered into position at the rate of 1 mm/minute (Exp1: AP +4.28 mm, ML +0.4

mm, DV 0.53 mm, Exp2: AP +4.28 mm, ML +0.5 mm, DV 1.0 mm). After reaching the target depth, the optetrode was adhered to the skull with C and B Metabond, followed by Teets 'Cold Cure' dental cement. After curing (10 min), the (op)tetrode was detached from the manipulator, the animal was removed from the stereotax and received subcutaneous injections of carprofen (10 mg/kg) and buprenorphine (0.05 mg/kg) and recovered on a heating pad kept at 37°C. The mice were monitored daily and received additional carprofen injections daily for the first two days postoperatively.

## Go-no go behavioral task

Mice were water deprived until they reached 80% normal body mass. Then they were placed into the Slotnick olfactometer (*Bodyak and Slotnick, 1999*; *Li et al., 2015*) chamber where they could move freely. All mice were first trained to lick the water spout to obtain water in the presence of odor (1% isoamyl acetate in mineral oil, v/v) in the 'begin' task (*Slotnick and Restrepo, 2005*). Subsequently they learned to discriminate 1% isoamyl acetate (S+) versus mineral oil (S-) in the 'go no-go' task (*Doucette et al., 2011*; *Li et al., 2015*), followed by learning to discriminate other odorant pairs (*Supplementary file 1*-Table S1). Data were analyzed for the go-no go discrimination task, but not for the begin task.

Mice self-initiated trials by poking their head into the odor delivery port, breaking a photodiode beam (*Figure 1A*). During reinforced odorant delivery (lasting 2.5 s) they must lick a water delivery spout at least once during each of four 0.5 second-long response areas in order to register the decision as a Hit (*Figure 1B*). Licks were detected as electrical connectivity between the water spout and the ground plate on which they stand (*Slotnick and Restrepo, 2005*). If the mice licked during a rewarded odorant trial, they received ~10 μl water reinforcement (*Figure 1C*). The mice learn to refrain from licking for the unrewarded odorant due to the unrewarded effort of sustained licking. For correct rejections mice leave the spout shortly after the last lick that takes place 0.5–1.8 s after odorant onset (*Figure 6B*). Performance was evaluated in blocks of 20 trials, with 10 S+ and 10 S- trials presented at random. Animals performed as many as 10 blocks per session. Sessions were terminated when animals demonstrated satiety/disengagement from the task or when they performed at or above 80% correct discrimination in three or more blocks in a session. For EAPAexp1, IAAPexp2 and EAPAexp2 once criterion was reached, the next session had reversed valence, meaning that the previous S+ became S- (and the previous S- became S+). This reversal disambiguates the identity of the odor from its valence. Data were analyzed for all odorant pair sessions, including isoamyl acetate vs. mineral oil. Data were analyzed within two performance windows: when the animal was performing below 65% (naïve) or above 80% (proficient). The data were also analyzed in the 65–80% performing window, and the results fell between the naïve and proficient windows (data not shown).

Odor stimulus delivery time was measured with a photoionization detector (miniPID, Aurora Scientific). *Figure 1—figure supplement 1* shows the time course for odorant concentration measured at the odor spout. The time difference between valve opening and detection of odor at the odor port was between 66–133 ms, depending on which olfactometer was used.

## Neural recording

Extracellular potentials from the four tetrodes were captured and digitized at 20 kHz on the RHD2216 amplifier of the Intan RHD2000 Evaluation System with a 1–750 Hz bandpass filter or amplified with a 16-channel amplifier (Model 3500; A-M systems; bandpass 1–5000 Hz), and sampled at 24 kHz by a DT3010 A/D card. Information on behavioral events (valve times, mouse presence at the odor port) was sent through a digital line from the Slotnick olfactometer to the Intan board. Licks detected by the Slotnick olfactometer were recorded as an analog signal by the Intan board.

## Phase Amplitude Coupling

PAC data were processed using the Hilbert transform method described by *Tort et al. (2010)*. Briefly, data were bandpass filtered with a 20th order Butterworth filter using Matlab's filtfilt function with zero phase shift to extract LFP in the low frequency oscillation used for phase estimation (theta, 2–14 Hz, *Figure 1Diii*) and the high frequency oscillation used for estimation of the amplitude of the envelope (either beta, 15–30 Hz, or high gamma, 65–95 Hz, *Figure 1v*). Hilbert transform established the theta phase (*Figure 1Div*) and, separately, the envelope for beta or high gamma (red line in *Figure 1Dv*). A plot of the distribution of beta/high gamma amplitude in 51 bins of the phase of

theta (PAC) was plotted to gauge the strength of PAC (*Figure 1E*). To quantify the strength of PAC we calculated the MI estimating the KL distance to quantify the difference between the observed beta/high gamma amplitude distribution along the phase of theta from a uniform distribution. If PAC is non-existent, MI = 0, meaning the mean amplitude is distributed uniformly over theta phases, and if PAC is a delta function MI = 1. MI for signals measured in brain areas such as the hippocampus typically fall between 0–0.03 (*Tort et al., 2010*).

## Phase-referenced LFP power

Beyond the mere existence of PAC that has been documented in numerous brain regions, we sought to evaluate whether the information carried by beta or high gamma power at different theta phases could be used to discriminate between olfactory stimuli. To accomplish this analysis, we developed the PRP approach using custom Matlab code. Briefly, PAC was calculated using the approach documented by *Tort et al. (2010)*, as described above and summarized in *Figure 1*. Peak and trough theta phases are defined as the phase for maxima and minima of the PAC distribution measured for the S+ trials (red plot in *Figure 2C*). A continuous Morlet wavelet transform was used to estimate the power for the high frequency oscillations (*Buonviso et al., 2003*). PRP was estimated as the power of the high frequency oscillations (beta or high gamma) measured at the peak or trough of PAC (*Figure 5*). The Matlab code used for data analysis has been deposited to https://github.com/restrepd/drgMaster (copy archived at https://github.com/elifesciences-publications/drgMaster-52583).

## PAC/PRP Simulation

To validate our analysis of PAC and PRP, we performed simulations with uncoupled oscillations (no PAC) with a low frequency cosine oscillation at 8 Hz and a high frequency cosine oscillation at 40 Hz (*Figure 1—figure supplement 2A*) or with coupled oscillations where high frequency bursts took place at 180 degrees for the low frequency cosine oscillation (*Figure 1—figure supplement 2B*). The bursts were generated by a cosine oscillation at the high frequency multiplied by gaussians with a full width half maximum of (1/8 Hz)/2 centered at 180 degrees of the low frequency cosine wave. Top panels in both S1A and S1B are pseudocolor Morlet wavelet power time courses, while the bottom panels show the PRP of the 40 Hz oscillation. For the case with no PAC (*Figure 1—figure supplement 2A*), peak and trough PRP are equal. However, for the PAC example (*Figure 1—figure supplement 2B*) the peak PRP is significantly higher than trough PRP.

## Statistical analysis

Statistical analysis was performed in Matlab. PAC parameters and PRP were calculated separately per electrode (16 electrodes per mouse) for all electrodes per mouse. Statistical significance for changes in measured parameters for factors such as learning and odorant identity was estimated using generalized linear model (GLM) analysis, with post-hoc tests for all data pairs corrected for multiple comparisons using false discovery rate (*Curran-Everett, 2000*). The post hoc comparisons between pairs of data were performed either with a t-test, or a ranksum test, depending on the result of an Anderson-Darling test of normality. GLM is a general statistical method that includes regression and analysis of variance. Degrees of freedom and statistical significance have the same meaning in GLM as in analysis of variance and regression (*Agresti, 2015*). In addition, as a complementary assessment of significant differences (*Halsey et al., 2015*) we display 95% confidence intervals (CIs) shown in the figures as vertical black lines were estimated by bootstrap analysis of the mean by sampling with replacement 1000 times.

## Linear discriminant analysis

Classification of trials using PRP was accomplished via LDA in Matlab whereby PRP for every trial except one were used to train the LDA, and the missing trial was classified by the LDA prediction. This was repeated for all trials and was performed separately for peak and trough PRP, and for analysis where the identity of the odorants was shuffled. LDA and dimensionality analysis were performed either on a per-mouse basis where the input was the PRP recorded from 16 electrodes, or on pooled mouse data where the input was the PRP recorded from 16 x N electrodes where N is the number of mice. For pooled mouse analysis a pooled response vector was therefore created by

concatenating across animals and the number of trials *n*, was determined by the session with the lowest number of trials for a single odorant (*Chu et al., 2016*).

## Dimensionality

Following *Litwin-Kumar et al. (2017)* we defined the dimension of the system (dim) with *M* inputs as the square of the sum of the eigenvalues of the covariance matrix of the measured PRP LFP (*pLFP*) divided by the sum of each eigenvalue squared:

$$\dim(pLFP) = \left(\sum_{i=1}^{M} \lambda_i\right)^2 \bigg/ \left(\sum_{i=1}^{M} \lambda_i^2\right)$$

where $\lambda_i$ are the eigenvalues of the covariance matrix of *pLFP* computed over the distribution of PRP LFP signals measured in the OB. If the components of *pLFP* are independent and have the same variance, all the eigenvalues are equal and dim(*pLFP*) = *M*. Conversely, if the *pLFP* components are correlated so that the data points are distributed equally in each dimension of an *m*-dimensional subspace of the full *M*-dimensional space, only *m* eigenvalues will be nonzero and dim(*pLFP*) = *m*.

## Acknowledgements

We thank Ms. Nicole Arevalo for animal husbandry, Ms. Dnate' Baxter for laboratory support, Dr. Douglas Curran-Everett for advice on statistical analysis, Dr. Cecilia Gauthier Umaña for discussions on wavelet analysis and Ms. Arianna Gentile-Polese for immunohistochemical assays. This research was supported by NIDCD 1F31DC016483-01A1 to JL and NIDCD 5R01DC000566-30 to DR.

## Additional information

### Funding

| Funder | Grant reference number | Author |
| --- | --- | --- |
| National Institute on Deafness and Other Communication Disorders | DC000566 | Diego Restrepo |
| National Institute on Deafness and Other Communication Disorders | DC016483 | Justin Losacco |

The funders had no role in study design, data collection and interpretation, or the decision to submit the work for publication.

### Author contributions

Justin Losacco, Conceptualization, Data curation, Software, Formal analysis, Validation, Investigation, Methodology; Daniel Ramirez-Gordillo, Conceptualization, Data curation, Formal analysis, Validation, Investigation, Methodology; Jesse Gilmer, Software; Diego Restrepo, Conceptualization, Data curation, Software, Formal analysis, Supervision, Funding acquisition, Project administration

### Author ORCIDs

Daniel Ramirez-Gordillo  https://orcid.org/0000-0002-8189-6069
Jesse Gilmer  https://orcid.org/0000-0001-5778-1061
Diego Restrepo  https://orcid.org/0000-0002-4972-446X

### Ethics

Animal experimentation: All animal procedures were performed under approval from the Institutional Animal Care and Use Committee (IACUC) at the University of Colorado Anschutz Medical Campus (protocol number 00270) under guidelines from the National Institutes of Health.

Decision letter and Author response
Decision letter https://doi.org/10.7554/eLife.52583.sa1
Author response https://doi.org/10.7554/eLife.52583.sa2

## Additional files

### Supplementary files

• Supplementary file 1. This file includes two tables: Table S1. Odorant pairs, implant device, mouse genotype and electrode locations for each experiment Odorant identities for each pair are listed along with concentrations represented as volume/volume dilutions in mineral oil (MO). In the case of EAPAexp1, EA is 0.1% ethyl acetate and PA is a mixture of 0.05% ethyl acetate + 0.05% propyl acetate. The two recording locations (Exp1 and Exp2) are listed in millimeters with respect to bregma. Note: Cre expression can affect physiological parameters and therefore, the differences found between experiments in this publication can be due to Cre expression (*Harno et al., 2013*). However, it is not evident to us which of the methodological differences between experiments contributes to statistical differences in measured parameters. Table S2. Total sessions per mouse, experiment and odorant pair. Odorant pair abbreviations derived from Table S1. The number of sessions are in the 'odorant pair' columns. Each row represents one animal.

• Transparent reporting form

### Data availability

The Matlab code used for data analysis has been deposited to https://github.com/restrepd/drgMaster (copy archived at https://github.com/elifesciences-publications/drgMaster-52583). Data have been deposited to GigaDB (https://doi.org/10.5524/100699).

The following dataset was generated:

| Author(s) | Year | Dataset title | Dataset URL | Database and Identifier |
|---|---|---|---|---|
| Losacco J, Ramirez-Gordillo D, Gilmer J, Restrepo D | 2020 | Supporting data for "Learning improves decoding of odor identity with phase-referenced oscillations in the olfactory bulb" | https://doi.org/10.5524/100699 | GigaScience Database, 10.5524/100699 |

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

**Appendix 1**

## Detailed evaluation of phase amplitude coupling on a per odorant pair basis

*Figure 2—figure supplement 2A and B* show the summary of changes with learning for S+ and S- odorants for MI (Ai, Bi) and the peak angle variance (Aii,Bii) for theta/beta (A) and theta/high gamma (B) for per electrode PAC for all odorant pairs. When evaluated on a per odorant pair basis theta/beta and theta/high gamma PAC strength, quantified as the MI calculated for LFPs recorded in each electrode, was stronger for recordings in Exp2 compared to Exp1, and there was a statistically significant increase in the MI for S+ with learning (naïve vs. proficient) (GLM analysis p value < 0.001, for event, proficiency and experiment, with the exception that p<0.01 for proficiency for theta/high gamma, 3255 d.f., number of mice is in *Supplementary file 1*-Table S1). Interestingly, there were decreases in peak angle variance for S+, and/or increases in peak angle variance for S- for theta/high gamma PAC when the animals became proficient (GLM analysis p values < 0.001 for event type, proficiency and experiment for both theta/beta and theta/high gamma, 3255 d.f., number of mice is in *Supplementary file 1*-Table S1).

## Comparison of decision-making times calculated using pooled animal PRP LDA data vs. per animal data

*Figure 6—figure supplement 1B and C* show that for both lick (B) and PRP peak LDA (C) the decision-making time is faster when data are pooled for all mice. Lick decision-making times were significantly faster with pooled data (GLM p value < 0.01, 21 d.f.) as were decision making times for peak PRP LDA (GLM p value < 0.01, 21 d.f.). Because pooled animal data yield faster decision times we did the comparison of decision times in *Figure 6* using pooled animal data.

