## [Decision Letter]

**Acceptance summary:**

For years, neuroscientists have hypothesized that coupling between the phase of one type of neuronal oscillation and the amplitude of another type of oscillation plays a role in learning. This paper presents new evidence to provide support for this hypothesis. The authors report that the theta phase-modulated amplitude of beta and gamma oscillations in the olfactory bulb can be used to discriminate between odorants after learning but not before. This study implies that certain aspects of oscillations that have been reported in memory systems of the brain (e.g., chunking of information within different phases of the theta oscillation in the hippocampus) also apply in the olfactory bulb, an early sensory structure and has implications for the universality of these principles throughout various systems of the brain. Scientists interested in learning and memory, the functions of oscillations, and neuronal ensemble coding of sensory stimuli will benefit from reading this work. An interesting future direction will be to assess the putative interaction between the effects reported here and sniffing patterns, considering that previous studies have shown that sniffing induces oscillations in the olfactory bulb and that there is an interaction between sniffing patterns and behavioral performance.

**Decision letter after peer review:**

Thank you for submitting your article "Learning improves decoding of odor identity with phase-referenced oscillations in the olfactory bulb" for consideration by *eLife*. Your article has been reviewed by Laura Colgin as the Senior Editor, a Reviewing Editor, and three reviewers. The following individuals involved in review of your submission have agreed to reveal their identity: Daniel W Wesson (Reviewer #2); Claire Martin (Reviewer #3).

The reviewers have discussed the reviews with one another and the Reviewing Editor has drafted this decision to help you prepare a revised submission.

Summary:

This study reports that theta phase-modulated amplitude of beta and gamma oscillations in the olfactory bulb can discriminate between odorants after learning but not before. These results have implications for how sensory-guided decision making may be orchestrated in the olfactory system. The hypothesis that the phase relationship between high frequency oscillations nested into lower frequency waves, or cross-frequency coupling, would carry coding information is very interesting and has been understudied in the olfactory system. Beyond applicability within the olfactory field itself, these studies hint that certain aspects of oscillations in more "central" established systems (like the chunking of information within different phases of the theta oscillation) also apply in the olfactory bulb, an early sensory structure, which has implications for the universality of these principles.

Essential revisions:

1) There is a lack of clarity in the presentation. Overall, the organization of the manuscript and the presentation of findings make it difficult for a broad readership to follow the logic of experiments and the analyses. The description of the results and figure panels jump around rather than flow through the figure. Further, many of the analyses lack sufficient motivation and interpretation within in the Results section. Thus, it is often difficult to understand why the analysis was done and what the outcome means. Revision of the manuscript should keep these general points in mind. Specific comments related to this general point are listed below:

a) Results section: There needs to be more description about the similarities and differences of Exp1 and Exp2 than just electrodes and recording depth. There is not enough information to determine if the results of both studies should be similarly interpreted. The findings in Exp2 appear to be stronger than Exp1, but this is not discussed further. The difference between Exp1 and Exp2 may arise because group 1 is composed of mice carrying Cre recombinase that is likely to change neurons physiology even in the absence of any recombination. This possibility should be mentioned in the Discussion section.

b) There are many times throughout the figures where findings are significant, but this is not indicated in the figure (i.e. with a symbol) and only stated in the figure caption or main text. It would be helpful if significant findings were presented uniformly throughout.

c) A more basic explanation of "dimensionality" and how this concept applies to the olfactory system would likely benefit readers who are unfamiliar with this concept. We suggest that some discussion of this in the Introduction as well as the Results section would be helpful.

d) From the start of the Results section, please mention the time window relative to poke from which data were analyzed.

e) Figure 1D: The data are presented like the x-axes are all the same and all data presented correspond to the expansion shown from Di to Dii. However, there are 16 cycles in Dii and Dv that don't line up and only 8 or 9 cycles in Diii and Div (these also don't appear to align). Did the timescale of the axis change?

f) Figure 2H: This figure is too small and difficult to work through. The main finding appears to be related to MI for S+>S- in trained animals across data sets. There is no consistent change in MI for S+ or S- with learning. There is a consistent increase in phase variance for S- with learning. These findings should be summarized somehow in a straightforward and easier to understand way.

g) Related to the above point, in Figure 2H, it seems that MI and peak variance evolution during learning are different for the test IAMO for both Exp1 and Exp2. This pair has the singularity of being between an odorant and the vehicle and not between 2 odorants. Could the difference be due to the fact that in this case the mouse has to detect the odorant but does not need to identify/discriminate it? Is the PAC different when the S+ is mineral oil compared to isoamyl acetate?

h) Figure 5, especially 5D, and its/their results are difficult to follow meaningfully. Please elaborate more upon what is being shown. Perhaps better defining "decision time" would serve well since it appears this is in reference to the LDA vs the animal's actual decision.

i) Subsection “Peak angle variance decreases for S+ and/or increases for S- when the animals learn to differentiate odorants in the go-no go task”: Make it clear that this is PAC for theta-gamma in this section. Overall, the treatment of theta-gamma versus theta-beta PAC is not consistent. The focus seems to be mainly on theta-gamma but occasionally beta is added in for comparison. Currently, this is more distracting than meaningful. PAC changes over learning seems different between theta/beta and theta/high gamma. Contrarily to the effect observed for high gamma (Figure 2 and Figure 2—figure supplement 1), it is interesting to see that mean MI for theta/beta (Figure 2Hi) increases with learning for both S+ and S-. How does one interpret this difference between the MI for theta/beta and theta/ high gamma relationship? Considering the two different sources (OB vs. feedback) of these oscillations, their potential phase differences with theta, and the capacity for population activity that supports them to be differentially processing information, it would strengthen the study to compare and discuss theta-gamma versus theta-beta PAC more concretely and clearly.

j) Subsection “Dimensionality of PRP LFP space decreases when the animal learns to discriminate the odorants” "evidence for a learned tightening of dimensionality for the odor representation manifold" Please clarify what is meant by odor representation manifold. This is interesting but not very clear.

k) In several other instances, the text/significance indicators on figures and the graphs themselves are quite small and difficult to see (e.g. Figure 3D-F).

2 Some parts of the methodology are not described in sufficient detail. Specific examples are as follows:a) The behavioral methodology is unclear with respect to the data that is analyzed. Is it that the animals are trained on the go-no go task with odor versus blank, then transitioned to S+/S-? Does learning correspond to the task part or just the new odor sets once the task is learned? The methods section is vague on these details.

b) All statistical comparisons are made using a GLM. This is not standard and requires more explanation and justification than provided. It would be helpful to state what significance and degrees of freedom mean in this respect.

3) It is frequently stated that decoding odor "identity" is enhanced with learning. It is not apparent from the data that this true. There is evidence that the contingency of an odor (rewarded S+ versus unrewarded S-) is decoded. The change in PRP upon odor presentation resembles a binary change in state (high for S+ and low for S-). However, it is not clear that this is truly odor "identity" specific. Specific points related to this concern are listed below:a) Would a decoder trained from data for one odor pair be proficient in decoding data from another odor pair in the same mouse? If so, the information carried is the correlation of the odor with reward (or not) rather than the specific odor itself. This would also be interesting but phrased differently.

b) It is unclear whether the PRP in proficient mice reflects the identity of the odorant rather than the fact that when smelling the odorant the animal is engaged in a focused activity if licking for a reward independently of the odorant identity, for the following reasons:i) According to the examples of S+ stimulations shown in Figure 1 and Figure 3, before the odorant onset, there is no strong oscillatory activity. The activity that start when mice are asked to give their answer (licking) seems to be long lasting. This activity lasts for seconds after the animal already gave his response to the stimulus (more than 2 seconds after odorant offset).

ii) The analysis encompasses a long time period (2.5 sec) during which animals display very different behavior between S+ and S- stimuli. Indeed, in the case of S+ stimulus, animals are actively engaged in the action of licking in order to receive the reward. This is very clear from the Figure 5A, B and D in proficient animals: the fact they start to actively lick as early as at 0.5 sec after the odorant onset show that they have already performed the discrimination. Whereas for S- stimulus, once they have identified the odorant, they can behave in many different ways, and I assume they are able to explore the room. How long did mice stay in the odorant port for no-go trials?

iii) The breathing during the focus action of licking in order to get the reward is very different from breathing during exploration. Can the licking-associated breathing by itself change the parameters measured in the study. In particular I am thinking of theta amplitude (Figure 2D) and frequency that could match the licking frequency. If the respiratory rhythm varies between S+ and S- trials, then it is likely to also change properties of high gamma rhythm and thus the PRP particularly for high gamma. Did you control for the breathing changes?

iv) The perfect correlation between peak PRP and decision making could also mean that PRP results from the action of licking vs. not licking in proficient mice and the fact that in proficient mice almost all the trials are correctly answered, limiting the variability.

4) It would strengthen the decoding and decision-making arguments if the linear decoding accuracy and behavioral performance per trial were correlated. Likewise, for LDA and lick timing on a trial-by-trial basis.

5) The authors clearly state the fact that the data were acquired from two separate experiments/groups of mice, each with different probe designs and different implantation coordinates. It's great that the authors are forthcoming with the fact that these experiments yielded different results on some occasions (i.e. higher PAC strength for theta/beta and theta/high gamma in the second experiment, as pointed out in Figure 2). What would be nice would be further consideration/discussion on possible reasons why these differences may have arisen. Similarly, some discussion of why different results were seen for different odor pairs is warranted.

6) It appears that theta-gamma PAC does not consistently change in for S+, although PRP does increase modestly. In contrast, it is potentially more interesting that PAC for S- may actually decrease over the course of learning (Figure 2—figure supplement 1) consistent with the increase in phase variance. This may be more closely related to the large decreases in PRP as well. Why wasn't this explored more? The emphasis seems to focus "increased" PAC and PRP for rewarded S+ (i.e. Discussion section).

7) The dimensionality reduction findings are not convincing and do not appear to add much to the study. Reduction is only significant when pooled across animals and Exp1 and Exp2. It is not clear how this should be interpreted.

8) Are beta and high gamma oscillations occurring at the same time during the course of the trial or is there an alternation of the 2 rhythms?

---

## [Author Response]

Essential revisions:1) There is a lack of clarity in the presentation. Overall, the organization of the manuscript and the presentation of findings make it difficult for a broad readership to follow the logic of experiments and the analyses. The description of the results and figure panels jump around rather than flow through the figure. Further, many of the analyses lack sufficient motivation and interpretation within in the Results section. Thus, it is often difficult to understand why the analysis was done and what the outcome means. Revision of the manuscript should keep these general points in mind.

We have revised the Results section to provide better motivation and interpretation of the analyses. In addition, we have modified the text and figures to provide more continuous flow through the figures.

Specific comments related to this general point are listed below:a) Results section: There needs to be more description about the similarities and differences of Exp1 and Exp2 than just electrodes and recording depth. There is not enough information to determine if the results of both studies should be similarly interpreted. The findings in Exp 2 appear to be stronger than Exp1, but this is not discussed further. The difference between Exp1 and Exp2 may arise because group 1 is composed of mice carrying Cre recombinase that is likely to change neurons physiology even in the absence of any recombination. This possibility should be mentioned in the discussion.

We appreciate the comment. The sentence in the Results section caused the reader to think that we thought that differences between experiments were due to electrode location. It was not our intent to do this. Indeed, genotype can influence physiological and behavioral results, but ground truth is neither C57BL/6, nor Cre mice. Importantly, the findings are statistically significant for both Exp1 and Exp2. Although the differences do tend to be larger for Exp2 it is our opinion that discussing whether this is due to the number of animals used, the location of the electrodes, the odorant pairs used or the genotypes is not granted because it would be too speculative. Instead, we state in the discussion that obtaining statistically significant results for both experiments, in the face of numerous experimental differences, is a strength of the study.

We have made the following changes to the text:

We have updated Supplementary file 1 to include the genotype per experiment and we have replaced the sentence with: “As shown in Table S1 the two experiments differed in electrode locations, mouse genotype, device implantation (tetrode vs. optetrode), and odorant pairs used (see also Methods).”

We have added a sentence to the Discussion section: “Importantly, our findings were replicated in two different experiments (Exp1 and Exp2) differing in genotype, electrode location, device implanted (tetrode or electrode) and odorant pairs used (see Table S1). […] Regardless of the reason for differences between experiments it is important for this study that the findings on PRP are found for both experiments under substantially different experimental conditions.”

We have added the following sentence to the legend for Supplementary file 1: “Note: Cre expression can affect physiological parameters and therefore, the differences found between experiments in this publication can be due to Cre expression (Harno et al., 2013). However, it is not evident to us which of the methodological differences between experiments contributes to statistical differences in measured parameters.”

b) There are many times throughout the figures where findings are significant, but this is not indicated in the figure (i.e. with a symbol) and only stated in the figure caption or main text. It would be helpful if significant findings were presented uniformly throughout.

We added asterisks to all bar graphs when we found significance in post hoc tests. We have not added significance asterisks to the cumulative probability histograms or time courses because this is awkward when you have several histograms in a plot. The significance p values for the cumulative probability histograms are in both the legend and the text in the results. In the literature we do not find asterisks for cumulative probability histograms. However, if the reviewers have a suggestion on how to do this, we will be glad to modify the plots.

c) A more basic explanation of "dimensionality" and how this concept applies to the olfactory system would likely benefit readers who are unfamiliar with this concept. We suggest that some discussion of this in the Introduction as well as the Results section would be helpful.

We have added an introduction of dimensionality in the Introduction and we have revised the Results section and the Discussion section making it clear to readers unfamiliar with this concept the definition of dimensionality and the relevance of the findings to olfactory signal processing (see essential revision point 7 below).

d) From the start of the Results section, please mention the time window relative to poke from which data were analyzed.

We have added this text at the start of the Results section:

“In the go-no go task mice start the trial spontaneously by poking their nose into the odor spout and licking on the lick port. The odorant is delivered at a random time 1-1.5 sec after nose poke.”

And this text was added at the start of the Results section:

“Finally, to evaluate the statistical significance of differences in oscillatory parameters estimated in this study the estimates were either averaged in the time period of odorant application (0.5 to 2.5 sec after diverting the odorant to the odor delivery spout) or the statistical significance was evaluated for the entire time course with time points every 0.1 sec using a generalized linear model (GLM, see Methods). Furthermore, we complement testing of statistical significance using p values with estimation of bootstrapped confidence intervals (Halsey et al., 2015).”

e) Figure 1D: The data are presented like the x-axes are all the same and all data presented correspond to the expansion shown from Di to Dii. However, there are 16 cycles in Dii and Dv that don't line up and only 8 or 9 cycles in Diii and Div (these also don't appear to align). Did the timescale of the axis change?

Thank you for noticing this. Yes, the expanded raw LFP (1-100 Hz) and the envelope for high gamma had been incorrectly extracted for two seconds. Figure 1 has been replaced including the correct traces.

f) Figure 2H: This figure is too small and difficult to work through. The main finding appears to be related to MI for S+>S- in trained animals across data sets. There is no consistent change in MI for S+ or S- with learning. There is a consistent increase in phase variance for S- with learning. These findings should be summarized somehow in a straightforward and easier to understand way.

We have replaced Figure 2H with an analysis on a per experiment basis where we use the average value of MI and peak angle variance for each odorant pair. This is a more conservative approach, more straightforward to present, that also finds a change in peak angle variance with learning using GLM. However, the more detailed analysis we had presented in Figure 2H is valid and provides additional relevant information. We have moved old Figure 2H to Figure 2—figure supplement 2 where it is shown as a full-page figure. We have moved the text describing Figure 2—figure supplement 2 to the Supplementary Information.

g) Related to the above point, in Figure 2H, it seems that MI and peak variance evolution during learning are different for the test IAMO for both Exp1 and Exp2. This pair has the singularity of being between an odorant and the vehicle and not between 2 odorants. Could the difference be due to the fact that in this case the mouse has to detect the odorant but does not need to identify/discriminate it? Is the PAC different when the S+ is mineral oil compared to isoamyl acetate?

Unfortunately, we do not have data where the S+ was mineral oil and the S- was isoamyl acetate. However, looking at the data in Figure 2—figure supplement 2 (former Figure 2H) we do not find the IAMO data particularly different from the other odorant pairs.

Indeed, asterisks indicate that for all the odorant pairs the differences are significant.

h) Figure 5, especially 5D, and its/their results are difficult to follow meaningfully. Please elaborate more upon what is being shown. Perhaps better defining "decision time" would serve well since it appears this is in reference to the LDA vs the animal's actual decision.

Yes, we are comparing the time when the animal decides to stop licking for S- (lick decision-making time) with the time when the difference between odorants using PRP LDA becomes statistically significant (PRP LDA decision-making time), and we show that these two times do not differ. We have thoroughly edited the subsection “Decision-making takes place at the same time for peak PRP LDA and lic”. This section now starts with a definition of what we mean with this decision-making time, and how it is calculated for licks and PRP LDA. In addition, we simplified Figure 6—figure supplement 1.

*i) Subsection “*Peak angle variance decreases for S+ and/or increases for S- when the animals learn to differentiate odorants in the go-no go task”*: Make it clear that this is PAC for theta-gamma in this section. Overall, the treatment of theta-gamma versus theta-beta PAC is not consistent. The focus seems to be mainly on theta-gamma but occasionally beta is added in for comparison. Currently, this is more distracting than meaningful. PAC changes over learning seems different between theta/beta and theta/high gamma. Contrarily to the effect observed for high gamma (Figure 2 and Figure 2—figure supplement 1), it is interesting to see that mean MI for theta/beta (Figure 2Hi) increases with learning for both S+ and S-. How does one interpret this difference between the MI for theta/beta and theta/ high gamma relationship? Considering the two different sources (OB vs. feedback) of these oscillations, their potential phase differences with theta, and the capacity for population activity that supports them to be differentially processing information, it would strengthen the study to compare and discuss theta-gamma versus theta-beta PAC more concretely and clearly.*

”Subsection “Peak angle variance for theta/high gamma phase amplitude coupling increases for S- when the animals learn to differentiate odorants in the go-no go task” We have made an explicit reference to theta-gamma in the opening of this section.

We have a very interesting finding about theta-beta PRP, that had not been given proper attention in the Discussion section (it was mentioned in part of a sentence buried within a discussion of theta/high gamma). We found that, in contrast to theta-gamma, theta-beta PRP does not differ between peak and trough. BETA LFP has been postulated to convey centrifugal modulation from downstream brain areas to the OB. Our findings suggest that centrifugal modulation does not differ through phases of the theta oscillation and would therefore excerpt modulation on high frequency circuit activity occurring at different theta phases. We have added a paragraph discussing theta/beta PRP. We did not focus on PAC differences between theta/beta and theta/gamma because these are not as clear as the peak/trough difference in PRP.

j) Subsection “Dimensionality of PRP LFP space decreases when the animal learns to discriminate the odorants” "evidence for a learned tightening of dimensionality for the odor representation manifold" Please clarify what is meant by odor representation manifold. This is interesting but not very clear.

We have revised the sentence as follows:

“This is evidence for a learned tightening of dimensionality for the odor representation manifold (defined as the independent neural components representing the odor).”

k) In several other instances, the text/significance indicators on figures and the graphs themselves are quite small and difficult to see (e.g. Figure 3D-F).

We have increased the size of the subset of the graphs that were too small, and we have increased the size of fonts.

2 Some parts of the methodology are not described in sufficient detail. Specific examples are as follows:a) The behavioral methodology is unclear with respect to the data that is analyzed. Is it that the animals are trained on the go-no go task with odor versus blank, then transitioned to S+/S-? Does learning correspond to the task part or just the new odor sets once the task is learned? The methods section is vague on these details.

We revised the Materials and methods section to make it clear that the data analysis was performed for sessions when the animal, well versed in the licking task, was learning to discriminate two stimuli. Thus, learning corresponds to learning to discriminate the two stimuli (go-no go task), as opposed to learning to lick to obtain water (begin task).

b) All statistical comparisons are made using a GLM. This is not standard and requires more explanation and justification than provided. It would be helpful to state what significance and degrees of freedom mean in this respect.

We consulted a biostatistician, Dr. Douglas Curran-Everett (Curran-Everett, 2000;

Halsey et al., 2015; Sharp et al., 2009), on the use of the generalized linear model (GLM). He indicated that our approach of using GLM followed by post-hoc t test/ranksum corrected for multiple comparisons using the false discovery rate is suitable for our multivariate analysis. He remarked that GLM is a general statistical method that includes regression and analysis of variance (ANOVA), methods that are more commonly used in publications, and that degrees of freedom and statistical significance have the same meaning in GLM as in analysis of variance and regression.

In addition, we would like to remark that in the figures we also display bootstrapped 95% confidence intervals that are recommended as complementary to the “fickle” p value in evaluation of significance (Halsey et al., 2015).

In the Materials and methods section we now state that:

“Statistical significance for changes in measured parameters for factors such as learning and odorant identity was estimated using generalized linear model (GLM) analysis, with post-hoc tests for all data pairs corrected for multiple comparisons using false discovery rate (Curran-Everett, 2000). […] In addition, as a complementary assessment of significant differences (Halsey et al., 2015) we display 95% confidence intervals (CIs) shown in the figures as vertical black lines were estimated by bootstrap analysis of the mean by sampling with replacement 1000 times.”

3) It is frequently stated that decoding odor "identity" is enhanced with learning. It is not apparent from the data that this true. There is evidence that the contingency of an odor (rewarded S+ versus unrewarded S-) is decoded. The change in PRP upon odor presentation resembles a binary change in state (high for S+ and low for S-). However, it is not clear that this is truly odor "identity" specific. Specific points related to this concern are listed below:

Correct, we did not mean to imply that the term identity was chemical identity of the odorant, but rather contextual identity (is the odor rewarded?). We have added these sentences at the beginning of the Results section to make this point clear:

“Importantly, reversal experiments from previous studies where we switched the rewarded and unrewarded odorants have shown that the power of the LFP in the OB and mitral/tufted cell spiking encodes for the *contextual* identity of the odorant (is the odorant rewarded?)(Doucette et al., 2011; Doucette and Restrepo, 2008; Li et al., 2015; Ramirez-Gordillo et al., 2018). Therefore, when we refer to identity in this publication, we do not mean the chemical identity of the odorant, we mean whether the odorant is rewarded or unrewarded.”

In addition, we now use the term “contextual identity” throughout, including the Abstract. Finally, we include a new Figure 4 described in subsection “Peak phase referenced power increases for S+ and decreases for S- over learning” showing that when odorant valence is reversed PRP switches indicating that PRP encodes for contextual identity.

a) Would a decoder trained from data for one odor pair be proficient in decoding data from another odor pair in the same mouse? If so, the information carried is the correlation of the odor with reward (or not) rather than the specific odor itself. This would also be interesting but phrased differently.

As shown in the new Figure 4 peak PRP switches when the rewarded odorant is reversed. Thus, peak PRP encodes for contextual identity (is this the rewarded odorant?). As explained in point 3 above, at the beginning of the results we make it clear that when we refer to identity we do not refer to the chemical identity of the odorant, we refer to contextual identity (is this the rewarded odorant?).

b) It is unclear whether the PRP in proficient mice reflects the identity of the odorant rather than the fact that when smelling the odorant the animal is engaged in a focused activity if licking for a reward independently of the odorant identity, for the following reasons:

As stated in point 3 above, we did not imply that PRP reflects the chemical identity. We have added two sentences to the Results section making this clear at the onset of the results.

i) According to the examples of S+ stimulations shown in Figure 1 and Figure 3, before the odorant onset, there is no strong oscillatory activity. The activity that start when mice are asked to give their answer (licking) seems to be long lasting. This activity lasts for seconds after the animal already gave his response to the stimulus (more than 2 seconds after odorant offset).

Correct, the activity lasts for a few seconds, and this raises the question whether this is an odorant response compared to reflecting centrifugal input to the OB. As stated above, our data indicate that PRP encodes for the contextual identity of the odorant.

However, we would like to remark that the long lasting time course of oscillatory activity does not necessarily imply that the response is not an odorant response because: (1) Our olfactometer is designed for fast odorant onset (with a time constant of 100 msec), but the decrease in odorant concentration takes place in a few seconds (see new Figure 1—figure supplement 1). (2) The odorant concentration is relatively high, several orders of magnitude above the threshold for detection.

ii) The analysis encompasses a long time period (2.5 sec) during which animals display very different behavior between S+ and S- stimuli. Indeed, in the case of S+ stimulus, animals are actively engaged in the action of licking in order to receive the reward. This is very clear from the Figure 5A, B and D in proficient animals: the fact they start to actively lick as early as at 0.5 sec after the odorant onset show that they have already performed the discrimination. Whereas for S- stimulus, once they have identified the odorant, they can behave in many different ways, and I assume they are able to explore the room. How long did mice stay in the odorant port for no-go trials?

For the S- mice left the spout shortly after the last lick. As shown in Figure 5B the last lick takes place at 0.5-1.8 sec after odorant onset. This is now sated in the Materials and methods section. We agree with the reviewer: in this go-no go task the animal is not motivated to leave the port immediately after detection of the S- odorant.

iii) The breathing during the focus action of licking in order to get the reward is very different from breathing during exploration. Can the licking-associated breathing by itself change the parameters measured in the study. In particular I am thinking of theta amplitude (Figure 2D) and frequency that could match the licking frequency. If the respiratory rhythm varies between S+ and S- trials, then it is likely to also change properties of high gamma rhythm and thus the PRP particularly for high gamma. Did you control for the breathing changes?

This is an interesting point. Active sniffing impacts odor responses through top-down and bottom-up mechanisms as shown, for example by Jordan and co-workers in a recent study (Jordan et al., 2018) and licks and sniffs are likely coordinated in the proficient animal given the fact that there is coordination of oromotor actions through brainstem circuits (Moore et al., 2014).

We did not measure breathing in this study. However, in past studies we have noticed close alignment of sniffs with theta LFP in the proficient animal in this go-no go task (Figure 5 of (Li et al., 2015)). With respect to the relationship of PRP power and licks, this is again an interesting question. Given the fact that there is coordination of oromotor actions through brainstem circuits (Moore et al., 2014) there is likely an alignment of licks, sniffs, whisker movement and theta LFP power when the animal becomes proficient. Indeed, studies in medial prefrontal cortex have shown that the LFP in the theta (6-12Hz) bandwidth is phase-locked to licks, even when the animal performs dry licks in the go-no go task (Amarante et al., 2017). Furthermore, in our past publication using Exp1 data, we performed a thorough analysis of event-related LFP power locked to the onset of the licks and we show that this measure changes as the animal learns and when we modulate noradrenergic input to the bulb through optogenetics (see Figure 7 in (Ramirez-Gordillo et al., 2018)). This published analysis of lick-related LFP power is directly relevant to the reviewer’s question on the relationship of licks and oscillations. In our opinion a more thorough study of the relationship of licks and oscillations is granted, but is beyond the scope of this manuscript.

iv) The perfect correlation between peak PRP and decision making could also mean that PRP results from the action of licking vs. not licking in proficient mice and the fact that in proficient mice almost all the trials are correctly answered, limiting the variability.

Yes, we agree that this can be the case (point iii above).

4) It would strengthen the decoding and decision making arguments if the linear decoding accuracy and behavioral performance per trial were correlated. Likewise, for LDA and lick timing on a trial-by-trial basis.

We have performed a trial-by-trial analysis of decoding performance and we have sorted the data according to the different behavioral outcomes. Figure 5E shows the time course for decoding performance analyzed on a trial-by-trial basis sorted by behavioral outcome (Hit, Miss, CR and FA, as well as shuffled). These results are presented in subsection “Linear discriminant analysis classifies stimuli using PRP”. Interestingly, shortly after addition of the odorant decoding performance decreases below 50% for FAs. This is further evidence that PRP does not encode chemical identity of the odorant.

We did not perform the lick/LDA analysis because we do not understand what the reviewers mean by a trial-by-trial “LDA and lick timing analysis”. In addition, we have published a study of lick-related LFP power using the data in Exp2 (Figure 7 of (Ramirez-Gordillo et al., 2018), see response to point 3.b.iii above). Furthermore, we would like to ask the reviewers to consider our opinion that, while interesting, a full study of the relation of licks to oscillations is beyond the scope of this manuscript (see response to point 3.b.iii above).

5) The authors clearly state the fact that the data were acquired from two separate experiments/groups of mice, each with different probe designs and different implantation coordinates. It's great that the authors are forthcoming with the fact that these experiments yielded different results on some occasions (i.e. higher PAC strength for theta/beta and theta/high gamma in the second experiment, as pointed out in Figure 2). What would be nice would be further consideration/discussion on possible reasons why these differences may have arisen. Similarly, some discussion of why different results were seen for different odor pairs is warranted.

Please see the answer to point 1a above. There are many differences between experiments: electrode locations, mouse genotype, device implantation (tetrode vs. optetrode), and odorant pairs used. Thus, while, for example, electrode location may explain the differences in measured variables, we think it is too speculative to suggest that the differences are due to one parameter. More importantly, the key findings are statistically significant for both experiments, and this is a strength of this study.

In the Discussion section we state: “Importantly, our findings were replicated in two different experiments (Exp1 and Exp2) differing in genotype, electrode location, device implanted (tetrode or electrode) and odorant pairs used (see Supplementary file 1). We did find differences in measured parameters between experiments, and these are likely due to methodological differences between Exp1 and Exp2 such as electrode location and genotype. Regardless of the reason for differences between experiments it is important for this study that the findings on PRP are found for both experiments under substantially different experimental conditions.”

6) It appears that theta-gamma PAC does not consistently change in for S+, although PRP does increase modestly. In contrast, it is potentially more interesting that PAC for S- may actually decrease over the course of learning (Figure 2—figure supplement 1) consistent with the increase in phase variance. This may be more closely related to the large decreases in PRP as well. Why wasn't this explored more? The emphasis seems to focus "increased" PAC and PRP for rewarded S+ (i.e. Discussion section).

We did not mean to focus on an increase in PRP for S+. Indeed, the sentence in the Discussion section also mentioned the decrease in PRP for S-. We have edited the sentence and performed a search/edit of the word “increase” in the manuscript. The sentence has been modified to: “We speculate that the concurrent increase in PRP for S+ and decrease in PRP for S- reflects adaptational mechanisms in the OB to enhance contrast between similar stimuli with different associated values.”

7) The dimensionality reduction findings are not convincing and do not appear to add much to the study. Reduction is only significant when pooled across animals and Exp1 and Exp2. It is not clear how this should be interpreted.

In the original version of the manuscript we did not present the results on dimensionality clearly, and there was a mistake on the statistical significance of the per experiment data. We have re-analyzed the data for both per mouse and pooled mouse data and we have generated new figures. As described in the revised Results section the increased decrease in dimensionality in proficient mice is significant for both per mouse and pooled animal data for both Exp1 and Exp2. This makes an important contribution to understand a sensory system that deals with high dimensional input because it indicates that learning focuses neural information processing on contextually relevant inputs. We have revised the Introduction and Discussion section to provide a clear presentation of the relevance of the analysis on dimensionality.

8) Are beta and high gamma oscillations occurring at the same time during the course of the trial or is there an alternation of the 2 rhythms?

Consistent with reports in the literature from, for example (Buonviso et al., 2003; Frederick et al., 2016), we did notice a slight delay of the increase in beta power compared to gamma. However, this delay was not as marked as reported in the literature and we did not focus on beta/gamma timing in this study.